# Fluid flow drives phenotypic heterogeneity in bacterial growth and adhesion on surfaces

Antoine Hubert[1], Hervé Tabuteau [2] ✉, Julien Farasin[1], Aleksandar Loncar [1], Alexis Dufresne[3], Yves Méheust[1] & Tanguy Le Borgne [1] ✉

Bacteria often thrive in surface-attached communities, where they can form biofilms affording them multiple advantages. In this sessile form, fluid flow is a key component of their environments, renewing nutrients and transporting metabolic products and signaling molecules. It also controls colonization patterns and growth rates on surfaces, through bacteria transport, attachment and detachment. However, the current understanding of bacterial growth on surfaces neglects the possibility that bacteria may modulate their division behavior as a response to flow. Here, we employed single-cell imaging in microfluidic experiments to demonstrate that attached *Escherichia coli* cells can enter a growth arrest state while simultaneously enhancing their adhesion underflow. Despite utilizing clonal populations, we observed a non-uniform response characterized by bistable dynamics, with co-existing subpopulations of non-dividing and actively dividing bacteria. As the proportion of non-dividing bacteria increased with the applied flow rate, it resulted in a reduction in the average growth rate of bacterial populations on flow-exposed surfaces. Dividing bacteria exhibited asymmetric attachment, whereas non-dividing counterparts adhered to the surface via both cell poles. Hence, this phenotypic diversity allows bacterial colonies to combine enhanced attachment with sustained growth, although at a reduced rate, which may be a significant advantage in fluctuating flow conditions.

Fluid flow is a common feature of bacterial habitats in soils, aquifers, rivers, and lakes, or in animal and plant bodies[1–7]. It modulates their chemical environment via the transport of nutrients, signaling molecules, and toxic compounds[8]. Flow near surfaces induces velocity gradients that can direct bacteria towards surfaces but also cause detachment of bound cells[4,9–13]. In mature biofilms, bacteria are protected from the direct mechanical action of flow by the Extracellular Polymeric Substances (EPS), and the interaction of bacterial colonies with the flow is mainly controlled by the EPS mechanical properties[14–19].

At the early stages of bacterial colonies, when the EPS matrix is not developed, flow modulates the spatial patterns and morphology of attached bacterial populations by physical processes, including bacterial transport, attachment, and detachment, which are considered to exert a strong influence on the future architecture of biofilms[20,21]. Bacteria can also sense flow by mechano-sensing[22–25] and modify the types of bonds to surfaces to enhance their adhesion depending on shear conditions[26–31]. However, it is not known whether such biological response may influence the colonization patterns and growth rates of bacterial colonies on surfaces exposed to flow.

Here we investigate the effect of mechanical stresses induced by fluid flow on bacteria division and attachment during early stages of surface colonization by a clonal population of *Escherichia coli*. We designed a microfluidic cell, allowing us to impose different magnitudes of shear stress while continuously providing nutrients and

[1]Géosciences Rennes, UMR 6118 University of Rennes and CNRS, Rennes, France. [2]Institut de Physique de Rennes, UMR 6251 University of Rennes and CNRS, Rennes, France. [3]ECOBIO, UMR 6553 University of Rennes and CNRS, Rennes, France. ✉e-mail: herve.tabuteau@univ-rennes.fr; tanguy.le-borgne@univ-rennes.fr

oxygen to a monolayer of Escherichia coli attached to the bottom surface of a flow channel. We used high-throughput tracking to monitor the motion and division of bacteria during a duration equal to 9 times the bacterial division time (i.e., ~ 6 hours). Our measurements reveal that flow induces an increase in phenotypic heterogeneity in bacterial division and attachment in clonal bacteria populations. Single-cell imaging shows that a sub-population of non-dividing bacteria coexists with dividing bacteria in attached colonies, with a proportion that increases with the imposed flow rate. Non-dividing bacteria were characterized by a strong adhesion to the substrate by their two poles while dividing bacteria were more asymmetrically attached. Hence, our findings demonstrate that clonal populations of bacteria can respond to flow by diversifying their growth and attachment phenotypes, which influences their colonization and growth rates on surfaces. By allowing a combination of dividing bacteria, vulnerable to erosion, and non-dividing bacteria, anchored to the surface, this strategy may be a key advantage for the resilience of microbial colonies subjected to variable flow conditions, a common situation that bacteria face in environmental and biological systems[4,32].

## Results

### Shear induces heterogeneous division rates in an isogenic bacteria population

Using microfluidic experiments, we explored the effect of fluid flow on bacterial growth and surface colonization by monitoring the rate of division, cell motion, and attachment and detachment ratio of bacteria exposed to different flow rates (see "Methods"). After performing pre-culture by diluting the stock culture in fresh medium and incubating it at 37 °C for ~7 h (see "Methods"), bacteria were injected in the chamber and let to sediment and attach to the bottom of the channel before experiments with different flow rates were started. The mean initial density of attached bacteria for all experiments was equal to about $10^{-2}$ cells per $\mu m^2$. The experiments were run under conditions of single-layer colonies (observation times less than six hours after which a second layer appeared). This allowed us to track all bacteria at a single-cell level throughout the experiments. For any of these bacterial trajectories, the tracking algorithm ended the trajectory at the time for which it was not able to unambiguously associate the bacterium's position in the previous image to a bacterium in the current image. This occurred either when a cell divided or when it detached. We differentiate these two types of events from the analysis of the mean square displacement of the bacteria's centers of mass at the end of the trajectory (see "Methods", Mean Square Displacement). For each shear rate, the experiments were performed in independent triplicate experiments with bacteria from different pre-cultures. In the following, we distinguish the doubling rate of a dividing bacterium and the growth rate of the population. The mean doubling time of a dividing bacteria is on the order of 40 min, independent of the applied flow rate. We define the average growth rate of the bacterial population on the surface as,

$$\eta = \left\langle \frac{1}{N}\frac{dN}{dt} \right\rangle, \qquad (1)$$

where $N$ is the instantaneous number of bacteria on the surface, and $\langle \cdot \rangle$ denotes a time average over the duration of the experiment. Since we perform our analysis in conditions where bacteria form a single-layer on the surface, $N$ is proportional to the total area occupied by bacteria on the surface. Measuring the average growth rate $\eta$ defined by equation (1) is equivalent to fitting and exponential growth $N = N_0 e^{\eta t}$ to the number of attached bacteria as a function of time. In contrast to the doubling rate, the growth rate of the attached population $\eta$ depends on the flow rate due to detachment, attachment, and phenotypic heterogeneity, as discussed in detail in the following.

Flow in the channel induced a shear rate $\dot{\gamma}$ ($s^{-1}$) at the surface where bacteria were attached:

$$\dot{\gamma} = \frac{\partial v}{\partial z}, \qquad (2)$$

which resulted in a shear stress $\tau_w$ (Pa),

$$\tau_w = \mu \frac{\partial v}{\partial z}, \qquad (3)$$

where $v$ is the fluid velocity in the channel, $z$ is the vertical coordinate and $\mu$ the dynamic viscosity of water. The shear stress values affecting attached bacteria were estimated by averaging the shear rates from $z = 0$ to $z = 3\,\mu m$, which corresponds to the average height of a monolayer of bacteria developing on the bottom of the channels (see Methods). The temperature of the setup was controlled to a value of 37 °C. The viscosity of water was, therefore, $\mu(37\,°C) = 0.691$ mPa.s[33]. We investigated four flow rates, corresponding to the different shear rates and stresses applied to attached bacteria (Table 1). To focus on the effect of shear only, we designed the experiment to ensure that the nutrient and gas fluxes were sufficiently large to avoid nutrient or oxygen limitations in all experiments (see "Methods").

The spatial patterns of colonization under the different flow rates followed those observed for other types of bacteria[21] (Fig. 1): a transition from uniform colonization at the ultra-low shear rate to a few clusters of bacteria randomly distributed at high shear rate. However, classifying bacteria according to their division time revealed an unexpected new element in these surface growth dynamics. For each observation time, we tagged bacteria attached to the surface according to the time at which they divided from the mother cell (see "Methods") in nine successive time intervals of forty minutes, which corresponds to the mean bacterial division time (Fig. 1 and Supplementary movies). We then quantified the population dynamics of each bacteria class under different flow rates (Fig. 2).

At ultra-low shear rate (ulow regime, Table 1), colonies grew until they almost fully covered the surface (Fig. 1a–c and Supplementary movie 1). Some colonies started to grow a second layer after six hours of growth and we thus analyzed bacteria dynamics up to that time. For each time interval, most bacteria attached to the surface at a given time had divided during one of the two previous time intervals, i.e., during the last 80 min. For instance, almost all bacteria present at $t = 360$ min (Fig. 1c) were divided between $t = 280$ and $t = 360$ min (red and orange colors). At a low shear rate (low regime, Table 1), bacterial colonies were more sparse although still fairly evenly distributed on the surface. The heterogeneity at the time at which the last division occurred increased significantly among attached bacteria (Fig. 1d–f and Supplementary Movie 2). At $t = 240$ min (Fig. 1e), recently divided bacteria (green colors) co-existed with a significant proportion of older bacteria that had not divided after $t = 120$ min (blue colors). Many of these old bacteria still persisted among recently divided bacteria (red and orange colors) at $t = 360$ min (Fig. 1f). At medium shear rate (med regime, Table 1) (Fig. 1j–l and Supplementary Movie 3), only a few

**Table 1 | Values of shear rate and shear stress for the investigated flow regimes, characterized respectively by: ultra-low (ulow), low (low), medium (med), and high (high) shear rates**

| regime | shear rate ($s^{-1}$) | shear stress (mPa) |
|---|---|---|
| ulow | 7 | 5 |
| low | 29 | 20 |
| med | 72 | 50 |
| high | 116 | 80 |

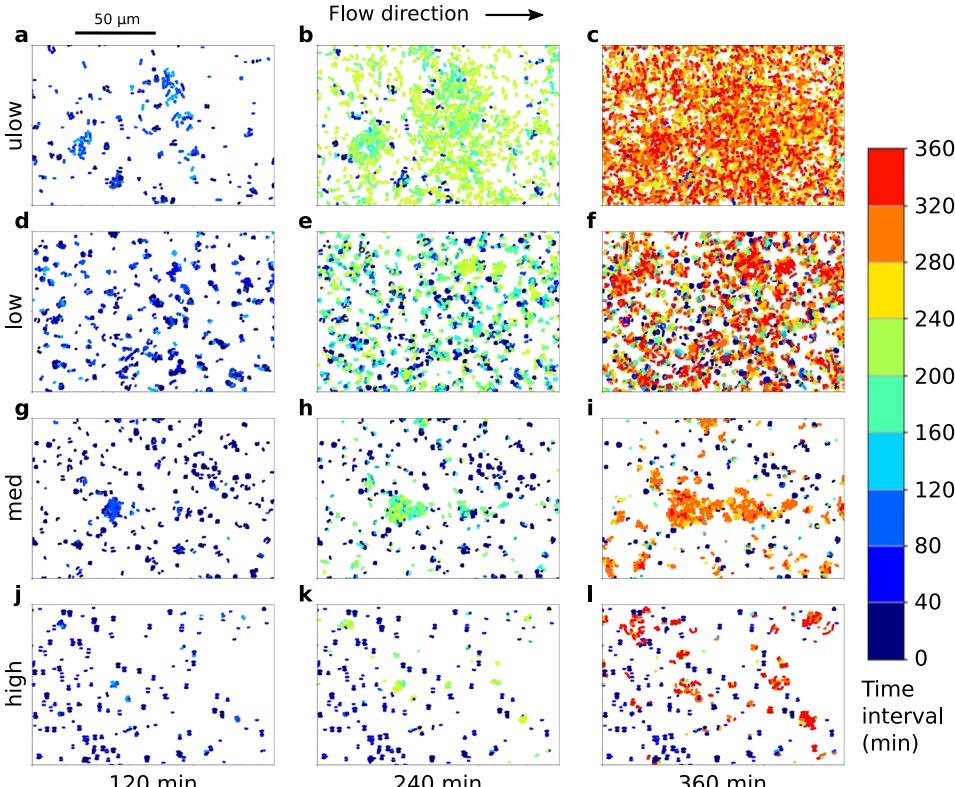

**Fig. 1 | Flow induces heterogeneous bacteria lifetimes between successive division events in bacterial colonies on surfaces. a–l** Maps of bacteria distribution classified according to their last division time in surface colonies of *Escherichia coli* bacteria exposed to flow. Snapshots of bacteria spatial distribution are shown in a section of the culture chamber for the four different regimes (Table 1): ulow (**a–c**), low (**d–f**), med (**g–i**), and high (**j–l**), at times 120 (left row), 240 (middle row) and 360 (right row) minutes after the start of the experiments. Dots represent bacterial cells. Colors correspond to the different time intervals during which a bacterium appeared on the image as a result of bacterial division. The birth time interval of initially attached bacteria is set to the first time interval, [0; 40] min, which is denoted by the dark blue color. The red color denotes bacteria formed during the 9th and last time interval. Fluid flow is from left to right. See Supplementary movies showing time-lapse images every 10 min.

colonies grew mostly along the flow direction, forming elongated patterns as colonies started merging with each other. The proportion of freshly formed bacteria at $t = 240$ min (green colors in Fig. 1k) and $t = 360$ min (red and orange colors in Fig. 1l) were much lower than for the ulow and low regimes and many old bacteria that had appeared in the first time intervals (blue colors) coexisted with the growing colonies. At the highest shear rate (high regime, Table 1), the proportion of old bacteria was larger, and only some patches of freshly divided bacteria could be observed (Fig. 1j–l and Supplementary Movie 4). Similar patterns were observed in independent triplicate experiments.

**Shear induced by flow can prevent bacteria from dividing**

Changes in the bacterial population growth were assessed under the different shear rates by measuring the ratio of the number of bacterial cells, $N$, to the value $N_0$ of that number at time 80 min, i.e., at the end of the second time interval (Fig. 2a–d). The population growth rate generally decreases with the imposed shear rate (Fig. 2a–d). The growth is approximately exponential for most of the observation times in all regimes, thus displaying a linear trend in semilog representation. This exponential trend develops after an initial lag time, discussed in more detail in the following. In the case of the ulow experiment, a significant reattachment of bacteria from upstream regions occurred after about 280 min, causing an acceleration of the attached population growth at late times. This was due to the formation of second layers of bacteria in upstream regions, which favored their detachment and downstream reattachment. We thus disregarded the late time data ($t > 280$ min) for this regime.

At any time, the global population dynamics may be decomposed into the growth of different bacteria classes by counting the number of bacteria that appeared during each of the successive time intervals. For each of them, $N/N_0$ increased to reach a maximum value and then decreased during the next time intervals when recently formed bacteria divided and were replaced by their daughter cells (colored curves). or the ulow and low regimes, the proportion of bacteria formed during the first time interval decreased rapidly, and these bacteria became a minority from the third time interval on ($t = 120 - 160$ min). For the med and high regimes, bacteria present in the first time interval were more persistent. They became less numerous than newly produced bacteria only after the 4th time interval ($t = 240 - 280$ min). In the high regime, a significant proportion of bacteria present in the first time interval had still not divided at the end of the experiment (see also Fig. 1l). Hence, the fraction of non-dividing cells increases with the shear rate.

**Erosion is not the unique factor limiting surface colonization underflow**

The reduction of population growth and colonization rate on surfaces exposed to flow has been observed in other studies[21,34,35] and explained by the balance between bacteria detachment and attachment. Our observations of bacterial division dynamics on surfaces suggest that bacteria can also respond directly to shear by stopping their division (Figs. 1 and 2). To determine the contribution of these different mechanisms, we quantified the detachment-attachment dynamics of bacteria as a function of the imposed shear rate. For this, we used high acquisition frame rate experiments (see "Methods"), to count the

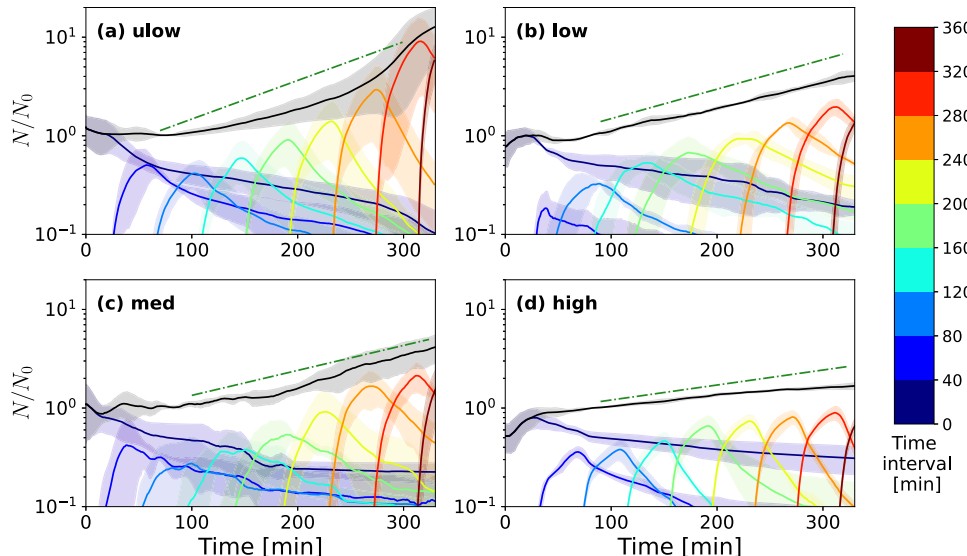

**Fig. 2 | Flow slows down the growth of bacterial populations on surfaces.** Growth dynamics of successive bacteria are classified according to the time at which they separated from the mother cell for the different flow regimes (Table 1): (**a**) ulow, (**b**) low, (**c**) med, and (**d**) high. The evolution of the total number of bacteria present on the surface (continuous black line), normalized by the number of bacteria $N_0$ at time 80 min (end of the second time interval), is decomposed into the different classes of bacteria defined according to the time intervals during which they appeared (colored lines). The dark blue color denotes bacteria present during the first time interval and the dark red color the most recently formed bacteria. The exponential trends shown as green dot-dashed straight lines correspond to the observed growth rates from Fig. 3. The average (solid lines) and confidence intervals (shaded areas) are estimated for each curve, respectively as the mean and standard deviations of bacteria numbers measured in independent triplicate experiments ($n$=3).

---

number of detachment and attachment events for each shear rate (Fig. 3a, b). We define the detachment ratios at a given time as the cumulative number of bacterial detachment events divided by the total number of attached bacteria measured since the start of the experiment

$$R_{\mathrm{d}}(t) = \frac{N_{\mathrm{d}}(t)}{N(t)} \qquad (4)$$

and the attachment ratio as the fraction of detached bacteria that reattached on the surface,

$$R_{\mathrm{a}}(t) = \frac{N_{\mathrm{a}}(t)}{N_{\mathrm{d}}(t)} \qquad (5)$$

The detachment ratio considered over the entire duration of the high acquisition frame rate experiments increased with shear from close to 0% in the ulow regime to about 50% in the high regime (Fig. 3a). Conversely, the attachment ratio (fraction of detached bacteria that reattached) decreased sharply with the shear rate. In the ulow regime, the vast majority of bacteria that detached from the surface reattached immediately after. These were individual cells which moved 5 to 10 μm away from their original location. In the low regime, ~70–80% of the detached bacteria reattached. This ratio dropped to 40% in the med regime, in which the bacteria reattached exclusively at the tail of the colonies along the flow direction. In the high regime, less than 10% of detached bacteria were able to reattach on the surface. This can be observed qualitatively in Fig. S3, where many bacteria appear in areas where there was no bacteria in the previous snapshot for the low flow regime (Fig. S3b), while only a few of the attachment events are observed in the high flow regime (Fig. S3k).

We define the effective population growth rate $\eta_{\mathrm{eff}}$ as the number of daughter cells newly formed by division (including cells that then detached from the surface) per unit of time, normalized by the number of attached bacteria. It is estimated from the observed growth rate of cells attached to the surface, $\eta$ (defined by Eq. (1)), corrected for the effect of detachment and attachment according to

$$\eta_{\mathrm{eff}} = \eta + \eta_{\mathrm{d}} - \eta_{\mathrm{a}}, \qquad (6)$$

where $\eta_{\mathrm{d}}$ is the detachment rate, and $\eta_{\mathrm{a}}$ is the attachment rate (Fig. 3c), both normalized by the number of attached bacteria $N$ and averaged over the experiment duration,

$$\eta_{\mathrm{d}} = \left\langle \frac{1}{N} \frac{dN_{\mathrm{d}}}{dt} \right\rangle, \qquad (7)$$

and

$$\eta_{\mathrm{a}} = \left\langle \frac{1}{N} \frac{dN_{\mathrm{a}}}{dt} \right\rangle. \qquad (8)$$

For the ulow and low regimes, the effective growth rate $\eta_{\mathrm{eff}}$ was approximately equal to the observed growth rate, $\eta_{\mathrm{eff}} \approx 0.54\,\mathrm{h}^{-1}$ for ulow and $\eta_{\mathrm{eff}} \approx 0.42\,\mathrm{h}^{-1}$ for low. For the med and high regimes, the observed and effective growth rates differed significantly. The observed growth rates $\eta$ were 0.35 and 0.21 $\mathrm{h}^{-1}$ for the med and high regimes, respectively, while the effective growth rates $\eta_{\mathrm{eff}}$ were 0.41 and 0.30 $\mathrm{h}^{-1}$ respectively. For the low regime, the decrease of the observed growth rate compared to the ulow regime ($-0.12\,\mathrm{h}^{-1}$) was partly due to the effect of erosion ($-0.02\,\mathrm{h}^{-1}$) but mostly attributed to lower production of cells by division ($-0.1\,\mathrm{h}^{-1}$). In the med regime, the observed growth rate decreased again but the effective growth rate remained similar to that in the low regime. This additional decrease was thus mostly due to enhanced erosion ($-0.07\,\mathrm{h}^{-1}$). In the med regime, the reduction of the observed growth rate compared to the ulow regime was $-0.33\,\mathrm{h}^{-1}$, with a contribution of one-third from erosion ($-0.1\,\mathrm{h}^{-1}$) and two-thirds from the reduction of bacterial division ($-0.23\,\mathrm{h}^{-1}$). As shear increased, the decrease of the average division rate of bacteria in response to flow thus became dominant over the effect of erosion. This observation is in contrast with the current

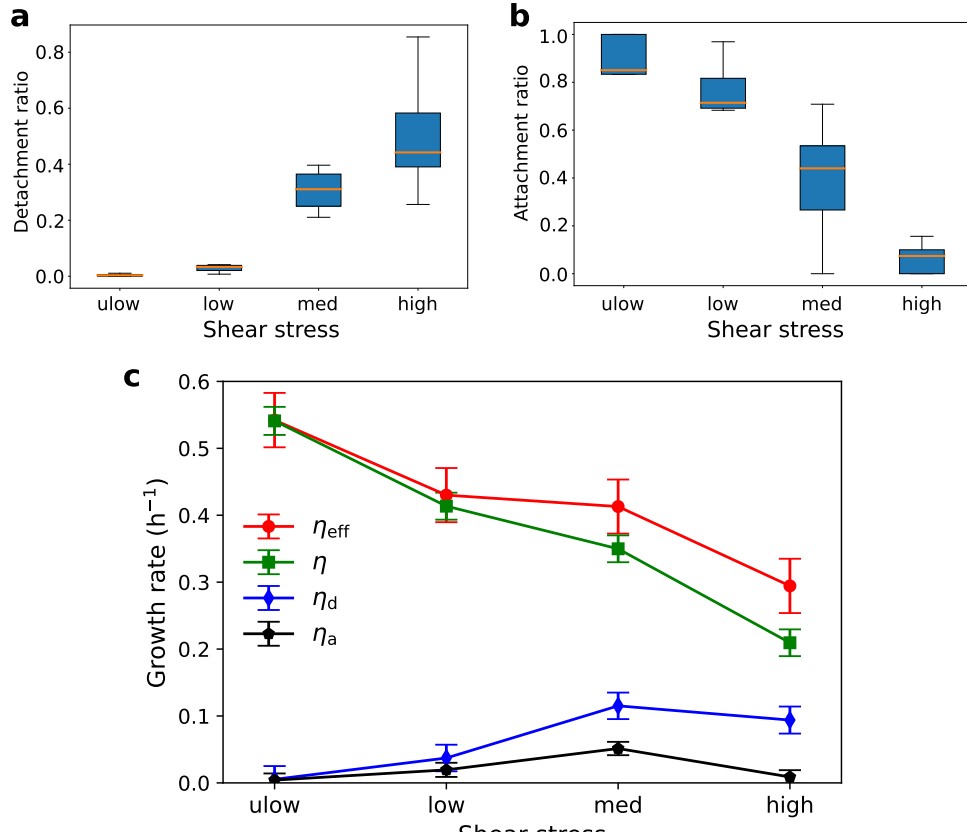

**Fig. 3 | Physical erosion does not explain the decay in bacterial growth rate with increasing shear. a** Ratio of the numbers of detached bacteria to attached bacteria for the different shear stress regimes (Table 1). The orange lines indicate the medians of the distributions. The boxes are centered on the mean and extend one standard deviation around it. The whiskers denote the first and third quartiles. **b** Ratio of the numbers of reattached bacteria to detached bacteria for the different shear stress regimes. The representation of statistics with box plots is the same as in subfigure (**a**). **c** Detachment rate ($\eta_d$), attachment rate ($\eta_a$), observed growth rate ($\eta$), and effective growth rate corrected for detachment and attachment, $\eta_{\text{eff}} = \eta + \eta_d - \eta_a$, for the different shear stress regimes. Even after correcting for detachment and attachment, the growth rates decay with the shear rate. The symbols denote the mean behavior, while the whiskers denote the $\pm 1$ standard deviation interval around the mean. In this figure, all data are obtained from the temporal statistics of similar experiments, as in Fig. 2 with a higher acquisition rate (10 frames per second). The sample size for these statistics (in bacteria numbers) is on average, $n = 1500$. The source data of this figure is provided in the source data file.

view, which considers erosion as the main mechanism responsible for limiting bacteria colonization of surfaces underflow [e.g., ref. 21].

**Fluid flow induces bistability in growth and attachment**

The analysis of the mean square displacement (MSD) of the bacteria's centers of mass (centroid) allowed tracking of the division events of attached bacteria at the single-cell level (Fig. 4a and "Methods"). Two groups of bacteria were identified based on the MSD of their centroid. Actively dividing bacteria, which we named dividers, were characterized by an average MSD on the order of $1\,\mu\text{m}^2$ at the end of their trajectory (when they became two separate bacteria). A second group, called non-dividers, had an MSD about two orders of magnitude smaller (Fig. S2), indicating that they did not divide during the observation time. Hence, two phenotypes coexisted in this isogenic population, and the spatial distribution of both groups was relatively uniform (Fig. S3a, d, g, j). The average fraction of dividers decreased with the intensity of shear (Fig. 4c), ranging from 80 to 63 percent for the lowest and highest shear, respectively. As the bacterial population grew with time, the number of non-dividers generally increased with time, indicating that a fraction of dividers continuously produced non-dividers (Fig. 4b). Hence, following the trend of the dividing population (blue curves on Fig. 4b), the number of non-dividers tended to increase faster in time when decreasing the shear (orange curves on Fig. 4b). However, the ratio of non-dividers over dividers (ratio of

orange to blue curves in Fig. 4b) increased with shear, consistent with the decay of the fraction of dividers (Fig. 4c).

The fraction of dividers $f$ and the effective growth rate $\mu_{\text{eff}}$ may be related by assuming that at each division event, a fraction $f$ of daughter cells remain dividers, while the other fraction, $1 - f$, become non-dividers. This leads to the recursive relationship $N_d(t + \overline{\tau_d}) = 2f N_d(t)$, where $N_d$ is the number of dividing bacteria and $\overline{\tau_d}$ is their average division time. The number of dividing bacteria hence evolves as $N_d(t) = (2f)^{t/\overline{\tau_d}}$, and the effective growth rate is:

$$\eta_{\text{eff}} = \frac{\ln 2f}{\overline{\tau_d}}. \tag{9}$$

This mathematical model is in relatively good agreement with the experimental data (Fig. 4c), when fitting the average division time to $\overline{\tau_d} \approx 52$ minutes. This value is similar to, although slightly larger than, the expected average division time (40 minutes). We discuss division time statistics in more detail in the following. Interestingly, the low and med regimes, which had similar effective growth rates, also had similar fractions of dividers. Equation (9) hence provides a link between the effective growth rate and the fraction of dividers, measured independently from each other, respectively from high frame rate imaging of bacterial motion (Fig. 3) and through MSD analysis performed on the lower frame rate measurements (Fig. 4b). It thus confirms that the

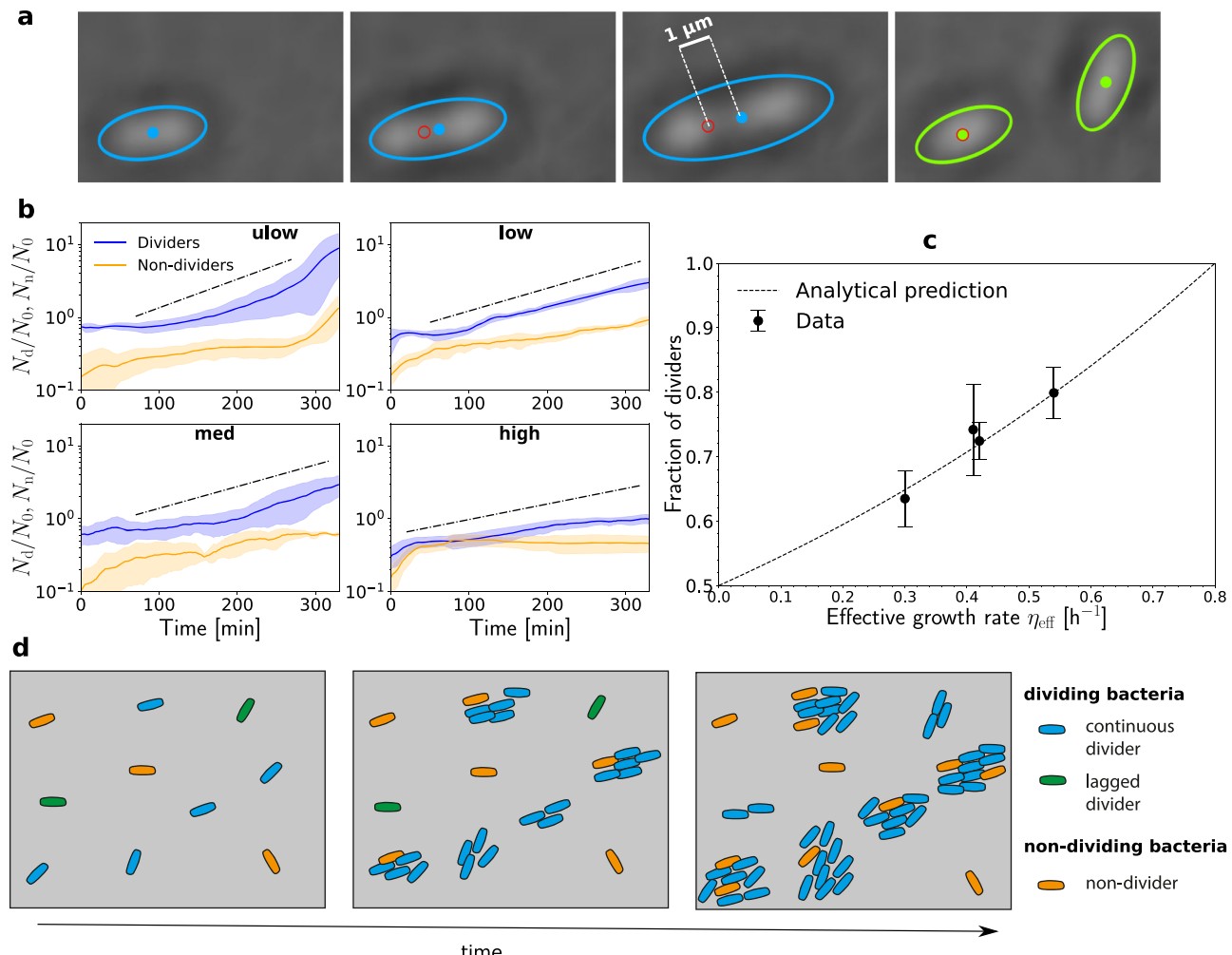

**Fig. 4 | The fraction of non-dividing bacteria increases with shear stress.**
**a** Division of a bacterium with fitted ellipsoid and centroid. During the division, the cell elongates in the direction of the flow. The initial position of the centroid is marked by the red circle. Such tracking of the bacteria's center of mass and size was performed in independent triplicate experiments for each flow regime. **b** Evolution of the normalized cumulative numbers of dividers $N_d$ (blue) and non-dividers $N_n$ (orange) in each flow regime (Table 1), normalized by the initial number of attached bacteria $N_0$. The average (solid lines) and confidence intervals (shaded areas) are estimated for each curve, respectively as the mean and standard deviations of bacteria numbers measured in independent triplicate experiments ($n = 3$). The exponential trends shown as black dot-dashed straight lines correspond to the effective growth rates from Fig. 3. **c** Average fraction of dividers as a function of the effective growth rates ($h^{-1}$). The dots represent the average fraction of dividers over the experimental time (Fig. 4b), and the error bars represent the standard deviation of the fraction of dividers estimated over the same time interval, with on average $n = 1500$ bacteria. The dashed line represents the model of equation (9) with $T = 52$ min. **d** Schematic representation of the different phenotypes of bacteria that develop underflow. Continuous dividers (blue) divide at the same average rate in all shear regimes. Lagged dividers (green) have an initial lag phase before they start growing at the same rate as dividers. Non-dividers do not divide at all over the period of observation. Dividers produce a fraction of non-dividers stochastically.

decay in the effective growth rate with flow may be explained by the increase of the fraction of non-dividing bacteria as a response to shear. We have summarized the bacterial growth behavior under shear in Fig. 4d.

## The division time statistics of dividing bacteria are independent of flow rate

To complement, the analysis of growth dynamics from the number of bacteria and their MSD, we analyzed the statistics of division times (Fig. 5a). These distributions were similar for all flow rates, indicating that the growth dynamics of dividers did not depend on shear. Hence, the observed dependence of growth rate on shear cannot be attributed to a change in the metabolism of dividing bacteria. This confirms the dominant role of the increase of the fraction of non-dividers with shear as discussed above.

In the low division time range, $20 < \tau_d < 160$ min, which included the large majority of measured division times, the distributions of division times were approximately exponential,

$$p(\tau_d) = \lambda\, e^{-\lambda \tau_d}. \tag{10}$$

This distribution corresponds to the expected Poisson process resulting from independent events occurring at a constant rate $\lambda$, were $\lambda^{-1} = 40$ min is the expected mean division time of the considered bacteria. In the large division time range (superior to 160 min), the division time distributions departed from the exponential distribution, particularly during the first time intervals. Hence, a small fraction of dividers had longer division times which were not captured by the exponential distribution (gray area in Fig. 5a), for all shear rates. We call this sub-group the lagged dividers and the other dividers the continuous dividers. We attribute the slower division of lagged dividers to a delay phase, followed by a division phase similar to that of the dividers (Fig. S4). The initial proportion of lagged dividers, between 10% and 15% of the global population, was similar for all shear rates.

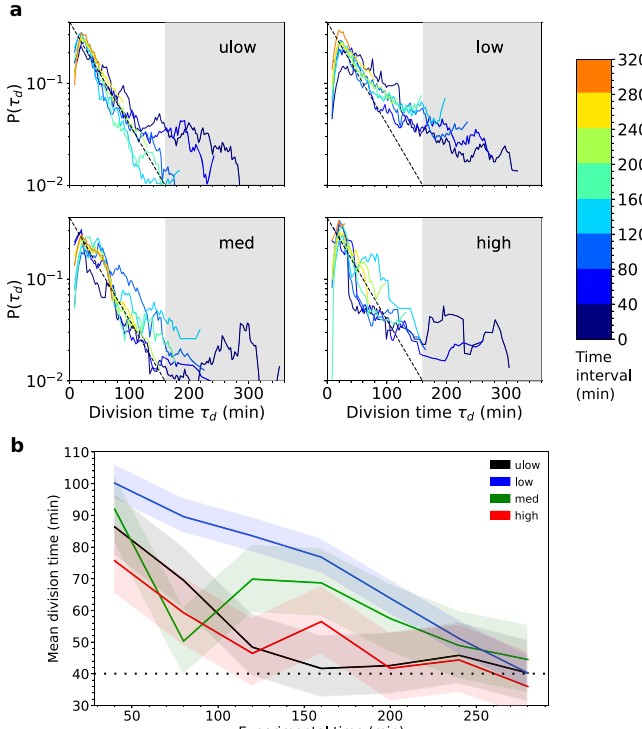

**Fig. 5 | Dividing bacteria maintain the same division time statistics in all shear regimes and accelerate their division rate in time. a** Probability Density Functions $P(\tau_d)$ of division times $\tau_d$ for each shear rate and all bacteria classes defined according to the time at which they appeared from the division of a mother cell (see color scale). The dashed line indicates the exponential distribution (Eq. (10)), corresponding to a Poisson process of rate $\lambda$, with $\lambda^{-1} = 40$ min. The gray area indicates the range of division times that are not captured by this tendency, which corresponds to lagged dividers. **b** Average division time as a function of observation time for the different shear rates. Standard deviations are indicated by the shaded color areas. For all shear rates, the average division time converges to $\lambda^{-1} = 40$ min (dotted line), as lagged dividers disappear in time. The sample size for these statistics (in bacteria numbers) is on average, $n = 1500$.

Because lagged dividers had much larger division times, they affected the initial average division time of the population, which ranged between 65 and 105 min (Fig. 5b). As lagged dividers underwent cell division, the distribution of division times tended to conform to the exponential distribution (Eq. (10)) and the mean division time of dividing bacteria decreased for all shear rates to reach a value of about 40 min, independent of the imposed shear (Fig. 5b).

### Non-dividing bacteria are more uniformly attached to the surface

The division of *Escherichia coli* is generally associated with an asymmetric adhesion to the surface by filamentous appendages that are located at cell poles[36]. The "old" pole, inherited from the mother cell, tends to be more strongly attached to the surface than the "new" pole formed at the division site. This can be tested by analyzing the mean square displacements (MSD) of the bacteria's cell poles. This metric is independent of the measurement of the MSD of the bacteria's center of mass, allowing us to characterize the symmetry of the bacteria's attachment in addition to their average mobility. For all shear rates, the old pole of continuous dividers moved about twice less than the new one (Fig. 6a), confirming their asymmetric attachment. For non-dividers, the MSD of their poles were similar and much smaller than those of continuous dividers for all shear rates (Fig. 6b). This suggests that the absence of division is associated with a more symmetric attachment to the substrate. For lagged dividers, the MSD of the two poles

were significantly larger than those of non-dividers (Fig. 6b). The magnitude of displacement of the two poles of lagged dividers was similar to the less mobile pole of the continuous dividers. We summarize the attachment modes of the three categories of bacteria in Fig. 6d suggested by the analysis of pole motion. The hypothesis proposed here of a link between growth arrest and symmetric attachment should be confirmed in the future by direct measurements of adhesion.

The more symmetric attachment of non-dividers suggests a stronger attachment to the surface. A further indication of this phenomenon is that the detachment rate (Eq. (7)) is slightly lower for the high shear than for the medium shear (Fig. 3c). This is possibly due to the decay in the fraction of dividers in the high shear regime compared to the medium shear regime (Fig. 4c). Since dividers are attached by one pole only, they are likely more easily detached than non-dividers, who are attached by both poles. Note that, despite this slightly smaller detachment rate, the detachment ratio (Eq. (4)) is larger for the high shear than the medium shear. For the high shear, bacteria from the few dividing colonies are continuously detached but the total number of bacteria on the surface, $N$ remains small due to the large fraction of non-dividers and the small reattachment probability (Fig. 3b). Hence the ratio of $N_d$ to $N$ is large. For the medium shear regime, the detachment rate is slightly larger (Fig. 3c) but about half of the detached bacteria reattach downstream (Fig. 3b) and can create new colonies. Hence, the number of attached bacteria grows more relative to the number of detached bacteria, as compared to the high shear regime. This leads to a smaller detachment ratio (Fig. 3a).

## Discussion

Phenotypic heterogeneity is a key component of the ability of bacterial populations to adapt and survive under environmental stresses, such as antibiotic and antiseptic treatment[37–43]. Our experimental results reveal that fluid flow leads to phenotypic heterogeneity in isogenic populations of *Escherichia coli*. We observed a significant reduction in the rate of surface colonization by bacteria when increasing the flow rate. This was partly due to the expected effect of physical erosion. However, a detailed analysis of bacterial division rates and motion on surfaces showed that a large part of the colonization slow-down had a biological origin as a large number of cells stopped growing and dividing. Thus, our experiments uncover an unexpected bacterial response to the physiological stress induced by flow.

Although populations of genetically identical cells were used in this study, the response to the shear stress induced by flow was not uniform. In the four flow regimes, we observed the coexistence of cells which divided actively during the experiment, maintaining the same average division time (~ 40 min), and cells which did not divide during the observation time (320 min). The two categories, the dividers, and non-dividers, were detected from the first time interval. The number of non-dividers generally increased in time (Fig. 4b), indicating that phenotypic heterogeneity was also present in new cells formed underflow. The proportion of non-dividers increased with the shear stress induced by the flow.

These different growth phenotypes were strongly correlated with attachment phenotypes. Non-dividers were characterized by their firm and symmetric adhesion to the substrate, while dividers displayed an asymmetric adhesion and a larger motion on the surface. Asymmetric adhesion is the dominant mode of attachment in *E.coli* bacteria colonies growing on surfaces[36]. However, the existence of phenotypic heterogeneity in attachment has also been observed in the absence of flow[44], with a fraction of symmetrically attached cells coexisting with asymmetrically attached cells. Growth rate diversity with the coexistence of dormant, slow- and fast-growing cells is also a common feature of isogenic bacterial populations cultured in static, homogeneous conditions[45,46]. Our findings suggest that phenotypic heterogeneity in surface adhesion and growth are tightly linked and

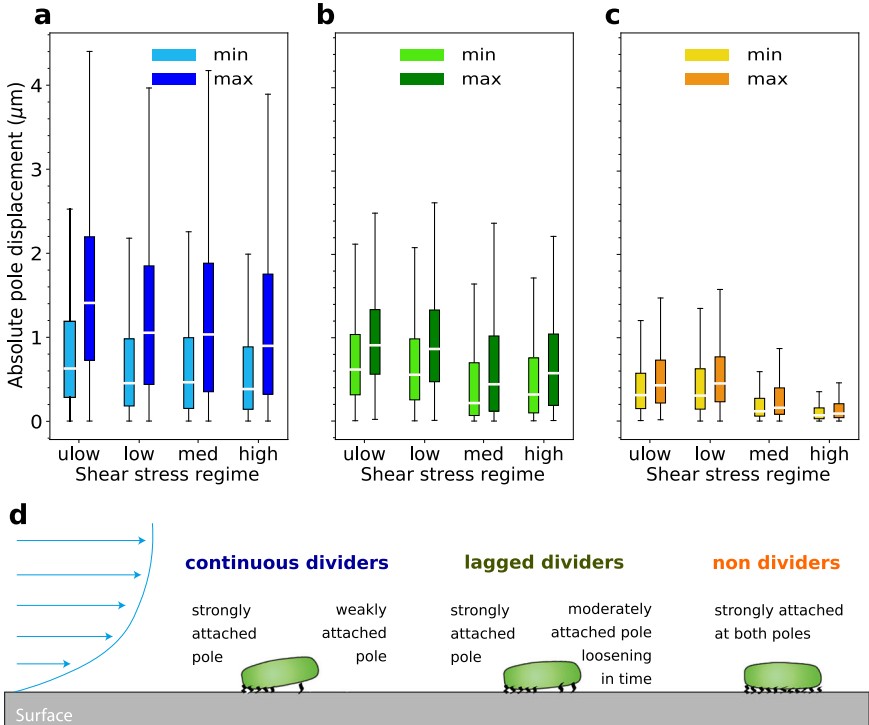

**Fig. 6 | Non-dividing bacteria are more symmetrically and more firmly attached.** Displacement of the bacteria poles for (**a**) continuous dividers, (**b**) lagged dividers, and (**c**) non dividers, for the different shear stress regimes (Table 1). For each bacterium, a pole with a larger displacement (max) and a pole with a smaller displacement (min) are identified. In this figure, all data are obtained from the temporal statistics of similar experiments as in Fig. 2. White central markers indicate the median displacement. The boxes extend to the first and third quartiles. The whiskers extend to one point five times the Inter Quartile Range (IQR). The sample sizes for these statistics (in bacteria numbers) are on average, $n = 1000$ for continuous dividers, $n = 400$ for non-dividers, and $n = 100$ for lagged dividers. **d** Schematic representation of adhesion modes for continuous dividers, lagged dividers, and non-dividers. The flow profile over the surface is represented in blue.

modulated by changes in hydrodynamic conditions, suggesting a force-sensing mechanism regulating cell attachment and growth rate.

The growth arrest observed here in non-dividing bacteria is comparable to the Viable But Non-Culturable (VBNC) state[47], a transient physiological state in which bacterial cells maintain viability but are not able to grow when cultured on non-discriminant media. It has been postulated that the VBNC state helps bacteria survive hostile conditions such as nutrient deprivation[48], UV exposure[49], or chlorination[50]. Cells in the VBNC state are also tolerant to antibiotics, and the VBNC state is considered to be related to the same dormancy phenotype as antibiotic persisters[51,52]. Although cell viability was not explicitly tested in this study, we assume that the non-dividing cells were still viable because displacements in their center of mass was observed during the experiments (Fig. 6c). In such laminar flows, dead cells would quickly reach a stable position with no detectable change in the MSD of their center of mass.

In Escherichia coli, one of the key molecules involved in the regulation of cell growth and stress response is the alarmone (p)ppGpp (guanosine tetraphosphate or guanosine pentaphosphate)[53,54]. A high concentration of (p)ppGpp in *E. coli* cells is associated with the induction of both VBNC and persisters states[55–58]. Interestingly, the concentration of this molecule also controls the production of type 1 fimbriae for bacterial attachment and biofilm formation[59,60]. We thus postulate that cell growth arrest and enhanced surface attachment may be two sides of the same bacterial response to flow-induced mechanical stress, possibly regulated by the same molecular mechanisms. Type 1 fimbriae could be involved in this process as a mechanosensor[25], possibly allowing force-induced feedback on bacterial adhesion. Increasing shear induces conformational changes in the structure of the FimH domain of type 1 fimbriae, therefore giving rise to a catch bonding mechanism that strengthens adhesion.

However, there is no demonstrated signal transduction mechanism associated with type 1 fimbriae that would allow bacteria to modify their cellular processes in response to changing shear conditions. This could be investigated in our setup by using a mutant that lacks type 1 fimbriae. However, type 1 fimbriae is necessary for irreversible adhesion of *E. coli* to surfaces[61–63], which is needed for *E. coli* to duplicate on surfaces. Hence, mutants lacking type 1 fimbriae do not grow on surfaces and their ability to form biofilm is considerably reduced[64,65]. The role of type 1 fimbriae in growth arrest should thus be confirmed in future studies, possibly by comparing gene expression and metabolic activity between non-dividing and dividing subpopulations of cells.

The uncovered phenotypic heterogeneity in growth rate and adhesion modes regulated by the flow intensity may be a type of bet-hedging, whereby genetically identical organisms develop heterogeneous phenotypes to prepare for an uncertain future[66–68]. This strategy provides bacteria with the ability to combine, at the population scale, cells that divide and are exposed to the risk of detachment with others that minimize the detachment risk by developing a strong attachment at the expense of immediate division. It is likely that bacteria have developed this strategy in environments where flow is highly fluctuating[38], such as soils or the gut[4,32]. Our preliminary experiments, including a sudden change from high to ulow (Fig. S5), suggest that bacteria adapt rapidly to a drop in flow rate by largely increasing their expansion rate. An interesting perspective of this study is, therefore to investigate how attached bacteria respond to fluctuations in flow conditions. These findings hence provide new insights on how bacteria manage the trade-off between division and attachment underflow, a key component to understanding the dynamics of bacterial growth and colonization in environmental, biological, and medical systems.

## Methods

### Bacterial strain and culture conditions

The *Escherichia coli* ATCC®11775™ strain was cultivated in 20 mL of M9 medium (per liter: Na$_2$HPO$_4$ 6g; KH$_2$PO$_4$ 3 g; NH$_4$Cl 1 g; NaCl 0.5 g; CaCl$_2$ 1M 30 µL; glucose 0.2 g/L) incubated at 37 °C in a 150 mL flask agitated at 150 rpm. A stock culture was grown from dehydrated discs in M9 minimal medium supplemented with glucose (2g.L-1) for 24 h to reach an O.D$_{600nm}$ of approximately 0.25. Under these conditions, the average bacteria size was 2 µm, as measured in minimal media[69]. The Stock culture was stored at 4 °C for a maximum of 3 weeks. Before each microfluidic experiment, a pre-culture was obtained by diluting the stock culture at 2% in fresh medium giving an O.D$_{600nm}$ of 0.005. The pre-culture was incubated at 37 °C for ~ 7 h, until reaching an O.D = 0.1. The evolution from OD = 0.005 to OD = 0.1 corresponds to 4 to 5 generations, and therefore all experiments were performed from pre-cultures in the mid-exponential growth regime. The pre-culture was diluted into fresh M9 medium to an O.D of 0.05 and transferred into a syringe (Cetoni GmbH glass syringe) before injection in the micro-fluidic device. The experiments were carried out in triplicate runs using different bacterial cultures to calculate error intervals.

### Microfluidic cell design and fabrication

The microfluidic device consisted of two superimposed channels, separated by a 150 µm thick PDMS membrane permeable to gas. Bacteria were grown in the bottom channel under fluid flow, while the top channel was flushed with air to ensure continuous delivery of oxygen to the bacteria cultures by gas diffusion through the PDMS membrane. The shear rate was varied by changing the flow rate in the culture channel. To estimate the shear rate and shear stress (Eqs. (2) and (3)) in the culture chamber, the velocity *v* was computed using the Stokes equation for Newtonian flow in a straight cuboid channel. The channels were 10 mm in length, 1 mm in width, and 150 µm in height. We fabricated microfluidic devices using soft lithography[70]. A poly-dimethylsiloxane (PDMS) mixture (Sylgard 184, Neyco s.a.) was poured into molds composed of embossed designs of SU-8 epoxy-based negative photoresist (SU-8 2050, Neyco s.a.) on silicium wafers (BT Electronics). The alignment of both PDMS layers was achieved with a magnifier trinocular zoom microscope (France-Tech Prochilab) using alignment patterns. The channels were bonded together and to a microscope glass slide covered by a thin layer of PDMS using a Corona SB for surface plasma treatment (BlackHole Lab).

### Environment control and experimental procedure

The bulk fluid and the solution containing bacteria were introduced into the system with syringe pumps neMESYS Low-Pressure modules 290N (Cetoni GmbH). The injection of gas in the gas control channels was performed with pressure controllers (25 mbar MFCS-EZ, Fluigent). The microfluidic chip was placed in a Leica Incubator 8 temperature control chamber, ensuring temperature stabilization at 37 ± 0.1 °C of the PDMS but also microfluidic tubings and circulating fluids. Bacteria were injected into the microfluidic cells and the flow was stopped for 30 min to let bacteria attach to the floor of the channels. Clean M9 medium supplemented with 2 g per liter of glucose was then injected from another syringe at the desired flow rate for 15 min before recordings were started.

### Nutrient and oxygen delivery

To focus on the effect of shear only, the nutrient and gas flux was set to be large enough to avoid any nutrient or oxygen limitations in all tested conditions. The characteristic consumption time of a dissolved species may be estimated as $\tau_c = ch/(\mu B)$, with *c* the species concentration, *h* the channel height, $\mu$ the consumption rate per cell and *B* the bacterial surface density. Oxygen transport is ensured by diffusion across the PDMS membrane and characterized by the diffusion time $\tau_D = e^2/D$, where *e* it the membrane width and *D* is the diffusion

coefficient of oxygen. Nutrients are delivered by flow in the chamber and characterized by the advection time $\tau_a = L/v$, with *L* the length of the chamber and *v* the flow velocity. The ratio of the transport time and the reaction time is thus characterized by the Damköhler numbers $Da^{O_2} = \tau_D/\tau_c^{O_2}$ for oxygen and $Da^n = \tau_a/\tau_c^n$ for nutrients. To estimate the nutrient Damköhler we focus on glucose since other elements were largely in excess. Typical oxygen and glucose consumption rates for *Escherichia coli* are respectively $\mu_{O_2} = 7 \times 10^{-21}$ kg.cell$^{-1}$.s$^{-1}$ and $\mu_{glu} = 5 \times 10^{-20}$ kg.cell$^{-1}$.s$^{-1}$ [71,72]. The characteristics of our experiments are: $c_{O_2} = 6.6 \times 10^{-3}$ kg.m$^{-3}$, $c_{glu} = 2 \times 10^{-3}$ kg.m$^{-3}$, $h = 1.5 \times 10^{-4}$ m, $L = 10^{-3}$ m, $10^{10} < B < 2 \times 10^{11}$ cell.m$^{-2}$, $4 \times 10^{-5} < v < 7 \times 10^{-4}$ m.s$^{-1}$. Therefore, we estimate $5 \times 10^{-4} < Da^{O_2} < 10^{-2}$ and $10^{-4} < Da^n < 3 \times 10^{-1}$ depending on the bacterial density and flow rate. This confirms that transport times are always smaller than the consumption time, implying that there were no oxygen or nutrient limitations in our experiments.

### Image acquisition

We used a motorized inverted microscope (DMi8, Leica Microsystems) to follow bacterial micro-colonies at the individual scale with an HC Plan 10x/25M ocular, an HCX PL Fluotar L 40x/0.60 CORR objective and an x1.6 tube lens (Leica Microsystems). Phase contrast images were acquired at a frame rate of 1 image per minute with an HPF-ORCA FLASH 4.0V3 camera (Hamamatsu). In order to obtain good statistics of bacterial counts, images were recomposed from 2 × 3 image mosaics acquired with the LAS X stitching module, and the best focus was guaranteed by performing a vertical scan over 8 µm around the initial best focus position, with a 0.5 µm interval with the LAS X Z-control module (Leica Microsystems). For the growth experiments, the size of images was 2048 × 2048 pixels, with a pixel size of 0.01 µm$^2$. For the high frame rate experiments (10 frames per second) used to quantify detachment/reattachment rates, the recorded area was a rectangle of length 651 µm and height 434 µm. In this configuration, the tube lens was replaced by a x1 tube lens (Leica Microsystems), and vertical stacks and horizontal stitching were not used. Hence, the recorded area was a square of side length 347 µm.

### Image processing and analysis

Recomposed images were pre-processed with in-house Matlab scripts for orientation correction, cropping, and best focus selection. Pre-processed images were then processed with an in-house Matlab program to identify individual bacteria and characterize their geometries (Fig. S1), see Supplementary Software). To this aim, the images were first treated through subtraction of a background image (obtained as the average of the 10 first images of the experiment, when very few bacteria are present in the system) and subsequent spatial filtering through a bandpass filter. The resulting images (Fig. S1) are decently contrasted, with a very uniform background intensity. Segmenting was then performed by using an intensity threshold, chosen depending on the gray level of the focused bacteria. The threshold value is fixed for any given experiment and chosen uniform over the entire image; but it differs between data sets. Using this procedure, few bacteria were left undetected even for a rather dense occupation of the surface. A sensitivity analysis for the efficiency of the segmentation was performed by changing the threshold by up to +/- 18 percent (which we consider a much larger uncertainty than that resulting from choosing the threshold manually based on visual impression). Changing its value within this large range led to a +/- 8 percent translation of the growth curve. Hence, once normalized by its reference value at time 100 min, the estimated growth rate indicated was independent of the chosen threshold value. These changes were negligible as compared to the variability from one data set to the other among the triplicates. At the x640 magnification with the 16-bit camera, each pixel covers an area of 0.01 µm$^2$. Considering *Escherichia coli* bacteria with a length of 2 µm and a diameter of up to 1 µm, each bacterium covers an area of ~200

pixels. This area depends on the actual position of the bacterium in the flow: if it is lying on the channel floor, it appears as rod-shaped on the image, while if it is attached by one pole and swung by the flow, it appears more round-shaped. Therefore, an additional selection was performed among the bacteria by applying a threshold on the eccentricities; we kept cells with eccentricity $e \in [0.5, 0.995]$. Note also that images were analyzed up to the time at which a second layer of bacteria started appearing. The second layer is not as well-focused as the first layer. Hence, bacteria from the second layer appear (i) larger on the image, (ii) of a lighter color tone, and (iii) blurry. These features allow us to determine the time at which a second layer of bacteria starts appearing. Individual bacteria were finally defined as connected white regions in the segmented images, using the Matlab library dedicated to this type of analyses, and their various geometrical properties (including their area, their center of mass position and that of their poles), were computed.

### Tracking of bacteria

Statistical data on bacteria's positions and geometric parameters as a function of time were then processed with an in-house Python script based on the *scipy* and *Trackpy* libraries to track particles in time[73], detect division events, and compute the mean square displacement (MSD, see below) along bacterial trajectories (see Supplementary Software). These data were analyzed using in-house Matlab scripts to: (i) partition the bacterial populations according to the 40 min time intervals of their birth, at all times; (ii) compute growth laws for the bacterial population according to their birth time intervals; (iii) compute the temporal evolution of MSDs; and (iv) distinguish dividers from non-dividers at all times and compute growth laws thereof.

At the considered bacteria densities, there were very few errors in bacteria detection by *Trackpy*. The absence of bias in the tracking statistics was confirmed by the statistics of bacterial division times, which are similar for all observation times (Fig. 5a). We did not observe any reduction of the bacterial track length due to segmentation or tracking errors as the density of bacteria increases. To further investigate the robustness of the tracking method, we have used high frame rate experiments in the dilute regime, where tracking errors with *Trackpy* are unlikely. The growth rates estimated from these high frame rate experiments (Fig. 3c) are compared to the growth rates of the standard experiments in Figs. 2 and 4b by plotting the corresponding exponential growth curves. In Fig. 2, we compare the observed rates of Fig. 3c, while in Fig. 4b we compare the effective rates of Fig. 3c. The good match in both cases confirms the lack of bias in bacteria tracking. From these high frame rate experiments, we have also shown that the attachment and detachment rates are generally much smaller than the division rate (Fig. 3). This implies that attachment/detachment does not significantly interfere with the estimation of division rates. The only exception is the ultra-low shear experiment, for which significant reattachment of bacteria from upstream regions occurs after about 280 min, causing an acceleration of the population growth at late times. This is due to the formation of second layers of bacteria in upstream regions, which favors detachment. We have thus excluded the late time data ($t > 280$ min) of the *ulow* experiments from the analysis.

### Detection of detachment and attachment events

To quantify the rates of detachment and attachment under different levels of applied shear, we performed experiments with a high frame rate, around 10 frames/second, allowing us to detect detachment events and follow the trajectories of bacteria after they had detached. The particles were tracked in time and whether they detach or attach was inferred from the time evolution of the distance between successive positions. The frequency of bacterial detachment and reattachment was estimated from three experiments performed at each shear rate and corresponding to a total of at least a hundred bacterial

detachment events over the 6 hour-long experimental period. Note that there is a certain amount of "flickering" in bacteria detection (see supplementary movies), that is, some particles are not visible at certain times. However, the *Trackpy* algorithm is able to reconstruct a trajectory even when a particle is not visible in several successive time frames within the trajectory, thanks to its prediction framework. Hence this does not affect the detection of attachment/detachment events.

### Mean square displacement (MSD)

The analysis of the mean square displacement (MSD) of the center of mass of bacteria, $\langle \Delta r^2 \rangle$, has been recently used to distinguish swimming, diffusing and anchored bacteria[44,74]. Here we used this measure to identify dividing and non-dividing bacteria. The identification of bacterial boundaries allowed us to estimate the position of the center of mass (centroid) for each bacterium in each image (i.e., at each time frame). As bacteria grew, we fitted their shape with an ellipse to track their growth during division and identify duplication events (Fig. 4a). During the duplication process, when the separation into two bacteria occurs, the trajectory of the centroid of the mother cell is lost, and two new trajectories appear, corresponding to the two new daughter cells. Hence the time over which the trajectory of a bacterium centroid can be tracked before separation, called here the lag time, corresponds to the time before the cell's division is complete. The *Trackpy* Python library which was then used to reconstruct bacterial trajectories $\mathbf{x}(t)$ (see above), also provided estimates of the MSD for a lag time $\tau$ as,

$$\langle \Delta r^2 \rangle(\tau) = \frac{1}{N} \sum_{i=1}^{N-k} \left( \mathbf{x}_{i+k} - \mathbf{x}_i \right)^2, \tag{11}$$

where $k = \tau/\Delta t$, $\Delta t$ is the time between successive positions recordings, and $N$ is the number of points in the trajectory, related to the trajectory's life duration $T$ by $T = N\Delta t$.

Once bacteria start dividing they also stop translating or rotating around significantly. Hence, during the division process, the variation in the MSD of a bacterium is mostly due to bacterial elongation. Attached bacteria grow until they have doubled their length before dividing (Fig. 4a). Since one of their pole is attached (Fig. 6a), their MSD increases by around half a bacterium's length, $\langle \Delta r^2 \rangle \approx 1 \mu m^2$, until the division is complete (Fig. S2). The evolution of the MSD is thus directly correlated to the division rate measured by the detection of division events. Every trajectory showing an MSD larger than $(0.5 \mu m)^2$ over the trajectory's duration (i.e., $\tau = T$) was thus considered to belong to the *divider* population (see blue curves in Fig. 4b). Among the trajectories showing an MSD smaller than the above threshold value over the trajectory's duration, those whose duration was smaller than the average bacterial division time of 40 min were not included in the non-dividers, as they could have detached before division. Those whose duration was larger than 40 min, on the contrary, were attributed to *non-dividers* (orange curves in Fig. 4b).

Note that the exponent of the MSD as a function of time has so far been used to distinguish swimming bacteria, which exhibit a ballistic behavior, diffusing bacteria, which show a diffusive behavior, and attached bacteria, which have a sub-diffusive behavior[44]. Here, the considered dividing and non-dividing bacteria are all attached to the surface. Hence, they generally show sub-diffusive dynamics for both classes, and the exponent of the MSD growth law does not provide a clear criterion to distinguish them.

### Relative pole displacement

We extended the MSD analysis to study the motion of bacterial pole movements to compare their level of attachment (Fig. 6). An ellipsoid was fitted to each bacterium and on each frame. From the fitted ellipsoid, two poles were identified. The evolution of the position of both poles and of the bacterial centroid over time yielded a total displacement for these three reference points. The ratio of each pole

displacement over the centroid displacement was thus obtained (see its statistics in Fig. 6).

### Initial attachment phase

In the initial attachment phase, the bacteria solution was injected into the microfluidic cell, and bacteria were left to attach at the bottom of the channel under no flow conditions for 30 min. After a brief sedimentation phase, bacteria started to adhere to the substrate. Observation of bacteria approaching the surface with the camera objective focused on the bottom of the channels showed that their mean velocity dropped from approximately $0.2\,\mu m.s^{-1}$ to $10^{-3}\,\mu m.s^{-1}$ once attached.

### Reporting summary

Further information on research design is available in the Nature Portfolio Reporting Summary linked to this article.

### Data availability

The main dataset used in this study is available on the zenodo repository: https://zenodo.org/records/11426128. Source data are provided with this paper.

### Code availability

The main codes used for data analysis are available on the Zenodo repository: https://zenodo.org/records/11426128. This includes: A Python code based on the Scipy and Trackpy libraries to track particles in time, detect division events, and compute the mean square displacement (MSD) along bacterial trajectories. A Matlab code to partition the bacterial populations according to the 40 minute time intervals of their birth, compute growth laws for the bacterial population according to their birth time intervals, compute the temporal evolution of MSDs, and distinguish dividers from non-dividers at all times. Additional codes performing secondary analysis are available upon request.

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

## Acknowledgements

This research was funded by the European Research Council (ERC) under the European Union's Horizon 2020 research and innovation program (Grant agreement No. 648377, T.L.B.). The funders had no role in the study design, data collection and analysis, decision to publish, or preparation of the manuscript.

## Author contributions

A.H. and H.T. designed and created the microfluidic devices. J.F. managed all microbiological elements. A.H. performed the experiments and contributed to data processing. T.L.B. and H.T. designed the research and contributed to data analysis and interpretation. A.L. contributed to data processing and analysis. Y.M. contributed to data processing, analysis, and interpretation. A.D. contributed to data analysis and interpretation. All authors contributed to manuscript writing and gave their final approval of the version to be published.

## Competing interests

The authors declare no competing interests.
