## [Peer Review File · Nature Communications]

Fluid flow drives phenotypic heterogeneity in bacterial growth and adhesion on surfacesReviewer #1 (Remarks to the Author):

The research described in the manuscript provides important evidence about the response of bacterial cells in biofilms to changing shear stress from the fluidic environment of the film. By using automated image analysis techniques to identify individual cells and cell events the observation populations are large enough to allow robust conclusions.

Clarification on the following points would be helpful.

1. Distinguishing between monolayer and multilayer growth: on page 4 you mention that a second layer growth occurs at later times. How is this kind of growth identified from the image analysis?

2. Creation of binary image (Figure SI.1b): was the intensity threshold established manually or was there an automated procedure for that step? Human-based thresholding is one source of trouble in reproducing the results of image analysis. When it is used it is important to show that the final results are independent of small changes in the threshold value.

3. Sample size for Figure 2: are the graphs of B/B_0 versus time shown in Figure 2 for a single experiment?

4. Meaning of attachment strength (p 8): on p8 there is discussion about attachment strength, in particular, it is stronger for non-dividers than for dividers. Can you be more explicit about the meaning of attachment strength? Since there are no actual measurements of the force required to remove a cell are you referring to lack of movements that reorient the cell?

5. Identification of a division event (Mean Square Displacement section on p13): this event is apparently identified by when a single cell trajectory bifurcates into two trajectories. What is the algorithm used by the software for determining this event? If it is explained explicitly in a reference then it would help to note it in this section.

6. Observation lag time (p14): How is this time defined?

7. Sample size for data shown in Figure 4 (p7): my reading of the text indicates that the uncertainty estimates shown in Figure 4b & 4c are based on the sample size of three separate experiments. If this isn't correct then it may be necessary to provide clarification.

8. Figure SI.4a confusion: I find Figure SI.4a a bit confusing. I assume the black lines correspond to size measurements of four individual cells as a function of time. Three of the cells show a time lag and one does not. Is the one cell with no time lag supposed to be a continuous divider? If so, I would probably remove it from this particular graph or make it clear that it corresponds to a different cell type. Also, are the blue dashed lines guides or actual fits to a model? I would suggest indicating the correct interpretation in the caption.

This was a really fascinating study.

Reviewer #2 (Remarks to the Author):

Description of the work:

Many studies have already reported that surface colonisation by bacteria under shear flow is dominated by attachment, detachment and erosion. In this manuscript, the authors want to demonstrate that surface colonization becomes less efficient at high shear rates because the growth rate of bacteria decreases as shear stress increases. Therefore, they performed microfluidic experiments, video microscopy and image analysis in order to measure the growth, attachment and detachment of bacteria at different shear rates.

General opinion:

It has been shown that bacterial growth is mechano-sensitive when bacteria grow in gels of different stiffness (Tuson et al., Mol. Microbiol. 2012). Here, the authors want to show that growth is also mechanical-sensitive under shear conditions. The impact of the study could be very large if there is indeed a mechano-sensitivity of bacterial growth to shear flow. However, the demonstration presented in this manuscript is based on indirect measurements, which suffer from two technical major issues that call into question the validity of the overall interpretation of the results: i) since bacteria are studied in very early exponential phase, growth heterogeneities might be a consequence of the growth phase rather than shear flow and ii) given the image quality and segmentation, it is not clear how the tracking algorithm behaves when the bacterial density is high. Therefore, if the authors want to demonstrate that bacterial growth is in fact dependent on shear stress, they should perform experiments on a mid-exponential culture and in a dilute regime where they can ensure that they are tracking single cells in order to provide the distribution of individual growth rates for different shear rates.

Major comments:

1. The bacteria culture protocol can introduce biases and heterogeneities in individual growth rates. As indicated in the Methods, the authors starts from a stock solution, which is a liquid culture in stationary phase stored for several days at 4°C. This mode of operation is not common practice in microbiology because bacteria experience very long stationary phase resulting in high mortality and very long lag phases before bacteria can resume to exponential growth. In addition, the longer the bacteria are stored in liquid conditions at 4°C, the greater these effects are. Therefore, if the different experiments corresponding to different shear rates are carried out successively on different days, the decrease in the effective growth rate will mainly reflect the fact that the culture is "ageing". As indicated in the Methods, bacteria from the stock solution are diluted at 2%, corresponding to an OD of 0.05 (if the OD at saturation of the stock is on the order of 2.5, which is a standard value). As the authors harvest the culture once it reached OD=0.1, it means that bacteria only grow for one generation, leaving a large part of the population in a lag phase. Different evidences in the manuscript tend to confirm that bacteria do not grow exponentially at the beginning of the experiment. Firstly, the length of E. Coli reported by the authors as 2µm is very short compared to normal conditions (4µm). Such short length is a hallmark of a stationary culture. Secondly, it seems from the Methods that bacteria need 7h to double from OD=0.05 to OD=0.1. Finally, the curves presented in Fig2 and Fig5b all show that growth becomes faster with time, indicating that bacteria are resuming to exponential growth during the course of the experiment (also illustrated by curves shown in FigSI-4). If the authors want to assert that bacterial growth depends on shear rates, they should make sure to start with a culture, which is in a steady-state exponential growth (so working with a mid-exponential phase OD=0.3). Moreover, the recipe for the M9 medium seems incomplete, as it is not mentioned that it is supplemented by a carbon source and amino-acids.

2. In shear flow experiments, the growth of the bacterial population is deduced from the dynamics of the number of bacteria detected (Fig2, Fig3). However, since bacteria from upstream regions can reattach in the field of view, it is very difficult to differentiate the contributions of division and reattachment. Trackpy is a Python library used to track the motion of individual particles in dilute colloidal suspensions. As with all tracking algorithms, its performances is considerably altered when the density of particles (here bacteria) increases. As the final density depends on the shear rate, the length of the tracks may become artificially shorter and new bacteria may appear artificially due to segmentation or tracking errors. In order to exclude possible artefacts due to image analysis errors and cell detachment and reattachment, the authors should first consider working in very dilute regimes and measuring the dynamics of the total biomass (for which the sum of all projected bacterial surface is a good proxy) over less than one generation in order to avoid crowding and reduce the probabilities of attachment and detachment.

3. Throughout the manuscript, there are some inconsistencies between the values of the growth rates and doubling times. In the text, on page 4 and 8, it is stated that the doubling time is 40-45min, while looking at Fig2 and Fig3, the doubling time appears to be on the order 80-100min. I guess the confusion is due to the fact that, as pointed in major point 1, the growth is not at a

steady-state during the experiment. Furthermore, how is that the doubling times can go down to 10 minutes (page 8 and FIG5a)? This very low value calls into question the fidelity of the tracking (see previous point).

4. The authors measured the MSD to show that non-dividing bacteria have symmetrical adhesion. However, the two features are both defined in relation to MSD : small for symmetrically attached bacteria and small as well if bacteria do not divide even if they adhere asymmetrically. Hence, on the basis of MSD alone, it is speculative to infer a correlation between non-dividing and symmetrically attached bacteria. However, the exponent of the MSD can be used to differentiate between growing from non-growing bacteria. Indeed, the center of mass of the growing bacteria will have a quasi-ballistic behaviour, while non-growing bacteria will be more diffusive.

5. The role of bacterial fimbriae is barely mentioned in the manuscript, although they have been shown to be sensitive to shear in the shear rate regime used in this study. The authors should consider conducting control experiments using a mutant of fimbriae production eliminate the mechanosensitive effects of bacterial adhesion.

6. It seems that only one biological replicate has been done for Fig 1 and 2.

7. The method of calculating the effective growth rate η_{eff} is not standard. Could the authors measure more direct properties such as cell elongation rate or total biomass growth rate by fitting exponentials?

Minor comments:

1. The term "generation" used in the manuscript is misleading. Figure 1 does not show the map of bacterial generations but rather the time when new tracks appear. Thus, a bacterium that has not yet divided during the 320 first minutes of the experiment but divides after 320 min will be labelled in red even if this is only the second generation.

2. Could the authors provide the raw movies of the experiments presented in Fig1?

3. The title of figure 1 is misleading because the color code does not directly reflect the measure of the division rate. Even if fewer bacteria divide at the beginning of the experiment, it is not clear if they divide more slowly.

4. On page 4, it is stated that "the heterogeneity in the temporal distribution of division events increased significantly" but there is no quantitative data to support this statement. Could the authors calculate the variance of these distributions?

5. On page 4, it is written that "the population growth followed a regular exponential trend (Fig2a)". This is not what I can see in Fig2a. Moreover, it would be clearer to plot these graphs in log-in scale rather than lin-lin.

6. The title of figure 2 is misleading, What is plotted are the growth curves, but not the growth rates calculated from exponential fits to these curves.

7. Could the author explain in the Methods how detachment and attachment events are computed? On the supplementary movies, it is clear that the images are imperfectly segmented and some bacteria disappear and reappear in successive frames. Could they also indicate these values in terms of events per bacteria and per unit of time?

8. A question somewhat related to minor point 6 concerns the details of the tracking parameters. Could the authors indicate in the Methods the distance that particles are allowed to move between two successive frames and whether the algorithm tolerates discontinuous trajectories in which bacteria disappear for one or more frames? In addition, the authors mentioned in the Methods that the length of a track reflects the division time. This is only the case if and only if there is no error on the track,

9. In figure 3 it would be more informative to use 2D scatter plots rather than bar plots. Indeed, since the value of the shear stress (or shear rate) is known for each condition, it is possible to plot the different rates as a function of the shear stress (or rate).

10. On page 6, it is written that "MSD on the order of $1\ \mu\text{m}^2$ ". Since MSDs are a function of time, could the author indicate that this value corresponds to the average MSD at division?

11. In the discussion, at the end of page 10, there are tons of scenarii other than stochasticity in gene expression that can account for heterogeneity.

12. The VBNC state is very different from the lag phase that bacteria experience when they resume from stationary to exponential phase. If bacteria are non-cultivable it is because we are unable to provide the right environment and sets of interactions for the bacteria to grow in a laboratory.

13. In the Methods, there is a typo in the M9 recipe. Replace, NCI by NH₄Cl.

14. Could the authors indicate in the Methods the time interval between frames for long experiments?

15. Equation 5 should be written in its discrete form.

Reviewer #3 (Remarks to the Author):

This is an interesting work which shows the response of *E. coli* cells to different flow strengths in terms of individual growth rates. Much effort has been devoted to image acquisition and image processing in order to identify heterogeneities in the behaviour of *E. coli* cells on the surface which may unveil previously unrecognized effects in bacterial surface colonization under fluid flow. However, I see some critical points that should be addressed by the authors.

First of all, the role of type 1 fimbriae - known to play a crucial role in the adhesion process to surfaces in *E. coli* - should be considered not only in the discussion section but also in terms of additional experiments with commonly available type 1 fimbriae mutant strains to confirm or reject the hypothesis that enhanced surface attachment and growth arrest are the two sides of bacterial response to flow.

As far as the heterogeneity is concerned, it would be interesting to see the effect of a rapid drop or a rapid increase of shear stress on the growth of bacteria on the surface: would non-dividers cells become dividers or viceversa?

Another crucial point is related to the statistical analysis: it is not clear to me if standard deviations correspond to population variability for a single experiment or if several experiments (i.e. different bacterial suspensions, different channels, etc.) have been performed.

Figure 3: data reported in the text for the observed and effective growth rates for the med and high regimes (0.4-0.48 and 0.25-0.35) do not correspond to the mean values shown in the graph.

REVIEWER COMMENTS

Reviewer #1 (Remarks to the Author):

The research described in the manuscript provides important evidence about the response of bacterial cells in biofilms to changing shear stress from the fluidic environment of the film. By using automated image analysis techniques to identify individual cells and cell events the observation populations are large enough to allow robust conclusions.

Clarification on the following points would be helpful.

1. Distinguishing between monolayer and multilayer growth: on page 4 you mention that a second layer growth occurs at later times. How is this kind of growth identified from the image analysis?

Response:

The second layer is not as well focused as the first layer. Hence, bacteria on the second layer appear (i) larger on the image, (ii) of a lighter color tone, and (iii) blurry (see Figure R1 below). These features allow us to determine the time in the experiment at which a second layer of bacteria starts appearing. We analyze experimental data only up to that time. **We have clarified this point in the revised manuscript (see Methods “Image processing and analysis” and page 16 in the annotated manuscript).**

Figure R1: Experimental image showing an example of a colony forming a second layer.

2. Creation of binary image (Figure SI.1b): was the intensity threshold established manually or was there an automated procedure for that step? Human-based thresholding is one source of trouble in reproducing the results of image analysis. When it is used it is important to show that the final results are independent of small changes in the threshold value.

Response:

The intensity threshold is fixed for any given experiment and chosen uniform over the entire image. The raw images are treated through subtraction of a background image (obtained as the average of the 10 first images of the experiment, when very few bacteria are present in the system) and subsequent spatial filtering through a bandpass filter. The resulting images (Figure R2(a)) are decently contrasted, with a very uniform background intensity, which allows using a single threshold value. The thresholds are chosen depending on the gray level of the focused bacteria. Fig. R2 below shows an example of thresholded image. The connected white clusters within the thresholded image are the red “blobs” shown in Figure R2(c). Each of these blobs corresponds to one or several bacteria. The number of bacteria it contains can be inferred from its size. On Figure R2(d) disks have been positioned at the center of mass of each detected blob; the color of each disk indicates how many bacteria it contains. Adding the numbers of bacteria for all blobs provides the entire bacterial count. Thus, as shown in Figure R2, even for a rather dense occupation of the surface by the attached bacteria in the first layer, few bacteria are left undetected.

Figure R2: (a) Raw image. (b) Thresholded image. (c) Detected blobs painted red on top of the original image. (d) Colored disks positioned at the center of mass of the blobs shown in (c); dark blue disks corresponds to blobs containing one bacterium, light blue disks to blobs containing two bacteria, green disks to blobs containing three bacteria, yellow disks to blobs containing four bacteria, the two red disks to blobs containing five bacteria, and the single pink disk corresponds to a blob containing six bacteria.

A sensitivity analysis for the efficiency of the thresholding was performed by changing the threshold by up to $\pm 18\%$ (which we consider a much larger uncertainty than that resulting from choosing the threshold manually based on visual impression). Changing its value within this large range led to a $\pm 8\%$ translation of the curve showing the number of bacteria as a function of time, so that, once normalized by its reference value at time 100 min, the growth rate indicated by the curve was independent of the chosen threshold value. These changes were negligible as compared to the variability from one data set to the other among the triplicates. This variability among the triplicates is shown by the confidence intervals in the new versions of Figures 2 and 3b, see as well our answer to point 3 below. Figure R3 below illustrates this dependence of the bacterial count on the threshold.

We have expanded the Methods section (“Image processing and analysis”) to explain this in more details (see page 16 in the annotated manuscript).

Figure R3: Dependence of the normalized bacteria count on the threshold chosen to detect bacteria in the videos.

3. Sample size for Figure 2: are the graphs of B/B_0 versus time shown in Figure 2 for a single experiment?

Response:

In the original manuscript, this graph of B/B_0 versus time in Figure 2 was made from a single experiment and the standard deviations were estimated from different images of a single experiment. In response to this comment, we have run a series of new experiments allowing us to define confidence intervals estimated as standard deviations over independent triplicate experiments (at each time). These new experiments confirm the trend of our original curves with relatively small confidence intervals. **We have clarified this in the revised manuscript in the paragraph “Shear induces heterogeneous division rates in an isogenic bacteria population” (page 2 in the annotated manuscript) and Methods section (“Bacterial strain and culture conditions”, page 15 in the annotated manuscript).**

4. Meaning of attachment strength (p 8): on p8 there is discussion about attachment strength, in particular, it is stronger for non-dividers than for dividers. Can you be more explicit about the meaning of attachment strength? Since there are no actual measurements of the force required to remove a cell are you referring to lack of movements that reorient the cells?

Response:

In this discussion about the attachment modes of dividers and non-dividers, we show that dividers tend to be attached asymmetrically, while non-dividers are generally attached equally by their two poles. This is shown by analyzing the relative motion of the two poles. To make this more explicit, we have removed the reference to the strength of attachment and replaced it by the asymmetric or symmetric nature of attachment.

5. Identification of a division event (Mean Square Displacement section on p13): this event is apparently identified by when a single cell trajectory bifurcates into two trajectories. What is the algorithm used by the software for determining this event? If it is explained explicitly in a reference then it would help to note it in this section.

Response:

We used the *Trackpy* Python library to reconstruct bacterial trajectories (e.g. Allan et al., 2015, Vaccari et al. 2018). In terms of particle tracking, a bacterium’s division results in two separate particles. Hence the original bacteria trajectory ends and two new trajectories appear. During division the length of bacteria increases and thus its center of mass moves (as one of its poles is firmly attached to the surface). We thus use a threshold in the value of the MSD at the end of trajectories to classify dividing bacteria and non-dividing bacteria. **We have clarified this in the Methods section (“Mean square displacement”, page 18 in the annotated manuscript).**

Allan, D, Caswell, T, Keim, N, and Wel, C van der. “Trackpy v0.3.2”. In: (2015). doi: 10.5281/zenodo.1213240.

Vaccari, L., Molaei, M., Leheny, R. L., & Stebe, K. J. (2018). Cargo carrying bacteria at interfaces. *Soft matter*, 14(27), 5643-5653.

6. Observation lag time (p14): How is this time defined?

Response:

The lag time is the time interval over which the MSD is computed along a bacterium’s trajectory. In other words, the MSD is a function of a lag time that varies from the minimum acquisition time to the lifetime of the trajectory. **We have clarified this in the Methods section (“Mean square displacement”, page 18 in the annotated manuscript).**

7. Sample size for data shown in Figure 4 (p7): my reading of the text indicates that the uncertainty estimates shown in Figure 4b & 4c are based on the sample size of three separate experiments. If this isn’t correct then it may be necessary to provide clarification.

Response:

The standard deviations in the original manuscript were estimated from different images of a single experiment. In response to this comment, we have run a series of new experiments allowing us to define confidence intervals from independent triplicate experiments. These new experiments confirm our original curves with relatively small confidence intervals. **We have clarified this in the revised manuscript in the figure captions and in the Methods section (“Bacterial strain and culture conditions”, page 15 in the annotated manuscript).** Note also that Figure 4b in its original form has been replaced by a new plot where the proportion of dividers is represented as a function of the observed growth rate of the entire population, providing a link with the independent high frame rate experiments of Figure 3.

8. Figure SI.4a confusion: I find Figure SI.4a a bit confusing. I assume the black lines correspond to size measurements of four individual cells as a function of time. Three of the cells show a time lag and one does not. Is the one cell with no time lag supposed to be a continuous divider? If so, I would probably remove it from this particular graph or make it clear that it corresponds to a different cell type. Also, are the blue dashed lines guides or actual fits to a model? I would suggest indicating the correct interpretation in the caption.

Response:

Yes, on this figure three curves correspond to lagged dividers and one is a continuous divider, included as a reference. The blue dashed lines are guides that indicate the mean growth rate of continuous dividers with no fitting. **We have clarified this in the figure caption (page 28 in the annotated manuscript).**

This was a really fascinating study.

Response:

We thank the reviewer for his/her constructive comments and enthusiastic evaluation.

Reviewer #2 (Remarks to the Author):

Description of the work:

Many studies have already reported that surface colonisation by bacteria under shear flow is dominated by attachment, detachment and erosion. In this manuscript, the authors want to demonstrate that surface colonization becomes less efficient at high shear rates because the growth rate of bacteria decreases as shear stress increases. Therefore, they performed microfluidic experiments, video microscopy and image analysis in order to measure the growth, attachment and detachment of bacteria at different shear rates.

General opinion:

It has been shown that bacterial growth is mechano-sensitive when bacteria grow in gels of different stiffness (Tuson et al., Mol. Microbiol. 2012). Here, the authors want to show that growth is also mechanical-sensitive under shear conditions. The impact of the study could be very large if there is indeed a mechano-sensitivity of bacterial growth to shear flow. However, the demonstration presented in this manuscript is based on indirect measurements, which suffer from two technical major issues that call into question the validity of the overall interpretation of the results: i) since bacteria are studied in very early exponential phase, growth heterogeneities might be a consequence of the growth phase rather than shear flow and ii) given the image quality and segmentation, it is not clear how the tracking algorithm behaves when the bacterial density is high. Therefore, if the authors want to demonstrate that bacterial growth is in fact dependent on shear stress, they should perform experiments on a mid-exponential culture and in a dilute regime where they can ensure that they are tracking single cells in order to provide the distribution of individual growth rates for different shear rates.

Response:

We thank the reviewer for his appreciation of the potential impact of our study and for her/his constructive criticisms, which have helped us clarify the description of our methodology. We have expanded our methods section to demonstrate that the two technical issues suggested by the reviewer do not apply to our study: i) the bacteria colonization dynamics are studied in the mid-exponential regime after 7 hours of pre-culture and not in the very early exponential phase, ii) data analysis is stopped when the bacterial density became too high to individually track them.

In the revised manuscript, we have included new data from independent experiments confirming the reproducibility of our results. In addition, we have expanded our analysis with a mathematical model (equation 4 and Fig. 4c) to demonstrate that the decay in population growth rate with flow, measured in high acquisition frequency experiments, is explained quantitatively by the increase in the proportion of non-dividing bacteria, measured in lower acquisition frequency experiments. The consistency between these independent experiments and the model confirms that the observed colonization and division patterns can be fully attributed to flow-induced shear

Major comments:

1. The bacteria culture protocol can introduce biases and heterogeneities in individual growth rates. As indicated in the Methods, the authors start from a stock solution, which is a liquid culture in stationary phase stored for several days at 4°C. This mode of operation is not common practice in microbiology because bacteria experience a very long stationary phase resulting in high mortality and very long lag phases before bacteria can resume to exponential growth. In addition, the longer the bacteria are stored in liquid conditions at 4°C, the greater these effects are. Therefore, if the different experiments corresponding to different shear rates are carried out successively on different days, the decrease in the effective growth rate will mainly reflect the fact that the culture is “ageing”. As indicated in the Methods, bacteria from the stock solution are diluted at 2%, corresponding to an OD of 0.05 (if the OD at saturation of the stock is on the order of 2.5, which is a standard value). As the authors harvest the culture once it reaches OD=0.1, it means that bacteria only grow for one generation, leaving a large part of the population in a lag phase.

Response:

It was perhaps not sufficiently clear in our original manuscript that we performed a pre-culture of 7 hours before injecting the bacteria in the microfluidic cells. This protocol directly addresses the concern of the reviewer by preventing the injection of an aging bacteria population. Our experiments were thus always performed in the mid-exponential growth regime, as recommended by the reviewer. We provide more details below and in the revised manuscript. Furthermore, we have also included new data from independent experiments performed at different dates for the same shear rates. The very good reproducibility of the experimental results excludes any possible aging effect.

Following standard practices (e.g. Sternberg et al. 1999, Secchi et al. 2020, or Barer et al. 2018), we have performed a pre-culture before each experiment by diluting the stock culture at 2% in fresh medium. The stock culture *Escherichia coli* ATCC®11775™ was grown from a dehydrated discs in M9 minimal medium supplemented with glucose (0.2 g.L⁻¹). After 24h growth at 37°C, the corresponding O.D. was approximately 0.25 (not 2.5 as estimated by the reviewer). Hence, dilution of the stock culture at 2% in fresh medium gave an OD of 0.005 (not 0.05 as estimated by the reviewer). The pre-culture was incubated at 37°C for approximately 7h, until reaching an O.D.=0.1. The evolution from an OD of 0.005 to an OD of 0.1 corresponds to 4 to 5 generations (not one as estimated by the reviewer) and therefore to the mid-exponential regime. The pre-culture is an important step, in which the optical density measurement allows standardizing both the physiological state and the density of the bacterial suspension injected in the microfluidic chip after dilution. It prevents injecting and aging bacteria suspension in the microfluidic chip, as suggested by the reviewer. **We have detailed more precisely this protocol in the Methods section (“Bacterial strain and culture conditions”, page 15 in the annotated manuscript).**

Different evidence in the manuscript tends to confirm that bacteria do not grow exponentially at the beginning of the experiment. Firstly, the length of E. Coli reported by the authors as 2 µm is very short compared to normal conditions (4µm). Such short length is a hallmark of a stationary culture.

Response:

The cell size of *E. coli* is a complex trait that depends on various factors, including nutrient availability (e.g. Yao et al. 2012, Hill et al. 2013) or stress (Dai et al. 2018). The length of 4 µm corresponds to *E. coli* in mid-exponential phase in rich media (such as LB for example). For other culture media, a smaller size has been measured for dividing *E. coli* cells in steady state exponential growth (e.g. Taheri-Araghi et al. in *Current Biology* (2015)). **We have commented this and added a reference in the revised manuscript in the Methods section (“Bacterial strain and culture conditions”, page 15 in the annotated manuscript).**

References:

Hill, N. S., Buske, P. J., Shi, Y., & Levin, P. A. (2013). A moonlighting enzyme links *Escherichia coli* cell size with central metabolism. *PLoS genetics*, 9(7), e1003663.

Yao, Z., Davis, R. M., Kishony, R., Kahne, D., & Ruiz, N. (2012). Regulation of cell size in response to nutrient availability by fatty acid biosynthesis in *Escherichia coli*. *Proceedings of the National Academy of Sciences*, 109(38), E2561-E2568.

Dai, X., & Zhu, M. (2018). High osmolarity modulates bacterial cell size through reducing initiation volume in *Escherichia coli*. *Mosphere*, 3(5), e00430-18.

Taheri-Araghi, S., Bradde, S., Sauls, J. T., Hill, N. S., Levin, P. A., Paulsson, J., ... & Jun, S. (2015). Cell-size control and homeostasis in bacteria. *Current biology*, 25(3), 385-391.

Secondly, it seems from the Methods that bacteria need 7h to double from OD=0.05 to OD=0.1.

Response:

Here, the confusion comes from the initial OD estimated by the reviewer to be 0.05 instead of 0.005. In 7h, the OD evolves from OD=0.005 to OD=0.1, which represents. 4 to 5 bacterial generations rather than one as estimated by the reviewer. **We have clarified this in the revised manuscript in the Methods section (“Bacterial strain and culture conditions”, page 15 in the annotated manuscript).**

Finally, the curves presented in Fig2 and Fig5b all show that growth becomes faster with time, indicating that bacteria are resuming to exponential growth during the course of the experiment (also illustrated by curves shown in FigSI-4). If the authors want to assert that bacterial growth depends on shear rates, they should make sure to start with a culture, which is in a steady-state exponential growth (so working with a mid-exponential phase $OD=0.3$).

Response:

As shown by the new version of Figure 2, which is presented in lin-log scale, growth is approximately exponential for all experiments, with a rate that is close to those measured precisely with the high acquisition frequency experiments (see green dot-dashed straight lines in Fig. 2). The acceleration of growth only concerns the ‘Ulow’ flow regime, and only after ~ 280 min. **We explain in the revised manuscript that this is attributed to a reattachment of bacteria from upstream regions (page 5 in the annotated manuscript).** This was due to the formation of second layers of bacteria in upstream regions, which favored their detachment and downstream reattachment. We thus disregarded the late time data ($t > 280$ min) for this regime.

As detailed above, all experiments were carried out with freshly prepared pre-cultures having reached the mid-exponential phase after 7 hours of growth (4 to 5 generations). The growth dynamics observed in our experiments are thus not due to any possible lag phase linked to the culture. **In the revised manuscript, we clarify this by providing more details on the experimental procedure in the Methods section (“Bacterial strain and culture conditions”, page 15 in the annotated manuscript).**

Moreover, the recipe for the M9 medium seems incomplete, as it is not mentioned that it is supplemented by a carbon source and amino-acids.

Response:

In the revised manuscript, we have provided more details on the recipe for the M9 medium in the Methods section of the revised manuscript (“Bacterial strain and culture condition”, page 15 in the annotated manuscript), and included in particular the carbon source concentration. There were no supplementary amino amino-acids in our growth medium.

2. In shear flow experiments, the growth of the bacterial population is deduced from the dynamics of the number of bacteria detected (Fig2, Fig3). However, since bacteria from upstream regions can reattach in the field of view, it is very difficult to differentiate the contributions of division and reattachment. *Trackpy* is a Python library used to track the motion of individual particles in dilute colloidal suspensions. As with all tracking algorithms, its performance is considerably altered when the density of particles (here bacteria) increases. As the final density depends on the shear rate, the length of the tracks may become artificially shorter and new bacteria may appear artificially due to segmentation or tracking errors. In order to exclude possible artifacts due to image analysis errors and cell detachment and reattachment, the authors should first consider working in very dilute regimes and measuring the dynamics of the total biomass (for which the sum of all projected bacterial surface is a good proxy) over less than one generation in order to avoid crowding and reduce the probabilities of attachment and detachment.

Response:

At the considered bacteria densities, there were very few errors in bacteria detection by *Trackpy* (see figure R4 showing an example of bacteria detection at one of the high densities). The absence of bias in the tracking statistics was confirmed by the statistics of bacteria division times, which are similar for all observation times (Fig. 5a). We did not observe any reduction of the bacterial track length due to segmentation or tracking errors as the density of bacteria increases. **We have commented this in the revised manuscript in the Methods section (“Tracking of bacteria”, page 17 in the annotated manuscript).**

Figure R4: Example of bacteria detection by *Trackpy* at one of the highest bacteria densities. Left: raw image; Right: detected bacteria (red).

To further investigate the robustness of the tracking method, we have used high frame rate experiments in the dilute regime, where tracking errors with *Trackpy* are unlikely. The growth rates estimated from these high frame rate experiments (Fig. 3c) are compared to the growth rates of the standard experiments in Fig. 2 and Fig. 4b by plotting the corresponding exponential growth curves; in Fig. 2 we compare the observed rates of Fig. 3c while in Fig. 4b we compare the effective rates of Fig. 3c. The good match in both cases confirms the lack of bias in bacteria tracking. From these high frame rate experiments, we have also shown that the attachment/detachment rate is generally much smaller than the division rate (Fig. 3). This implies that attachment/detachment does not significantly interfere with the estimation of division rates. The only exception is the ultra low shear experiment for which significant reattachment of bacteria from upstream regions occurs after about 300 minutes, causing an acceleration of the population growth at late times. This is due to the formation of second layers of bacteria in upstream regions, which favors detachment. This only influences the late time growth of the lowest flow experiment and does not affect our general analysis and conclusions. **We have commented this in the revised manuscript in the Methods section (“Tracking of bacteria”, page 17 in the annotated manuscript).**

3. Throughout the manuscript, there are some inconsistencies between the values of the growth rates and doubling times. In the text, on page 4 and 8, it is stated that the doubling time is 40-45 min, while looking at Fig2 and Fig3, the doubling time appears to be on the order 80-100 min. I guess the confusion is due to the fact that, as pointed in major point 1, the growth is not at a steady-state during the experiment. Furthermore, how is that the doubling times can go down to 10 minutes (page 8 and Fig5a)? This very low value calls into question the fidelity of the tracking (see previous point).

Response:

It is important here to distinguish the doubling time of a dividing bacterium and the growth rate of the population. The doubling time is indeed on the order of 40 minutes, as stated consistently in the manuscript and demonstrated in Figure 5. However, the growth rate of the entire bacterial population is smaller because of the presence of non-dividers and lagged dividers in the population. To clarify this, we have included a mathematical model relating the effective growth rate to the fraction of non-dividers (equation 4 in the revised manuscript). The model agrees well with the data obtained from independent experiments (Fig. 4c). **We have discussed this in the revised manuscript in the paragraphs “Shear induces heterogeneous division rates in an isogenic bacteria population” (page 2 in the annotated manuscript) and “Fluid flow induces bistability in growth and attachment (page 8 and 9 in the annotated manuscript).**

As shown in Figure 5, division events of 10 minutes are very scarce and most dividing bacteria have a division time in the interval 20-160 minutes. The division statistics follow a well defined exponential distribution, consistent with a constant rate random division process. **We have commented this in the revised manuscript in the section “The division statistics of dividing bacteria are independent on flow” (page 9-10 in the annotated manuscript).**

4. The authors measured the MSD to show that non-dividing bacteria have symmetrical adhesion. However, the two features are both defined in relation to the MSD : small for symmetrically attached bacteria and small as well if bacteria do not divide even if they adhere asymmetrically. Hence, on the

basis of the MSD alone, it is speculative to infer a correlation between non-dividing and symmetrically attached bacteria. However, the exponent of the MSD can be used to differentiate between growing from non-growing bacteria. Indeed, the center of mass of the growing bacteria will have a quasi-ballistic behaviour, while non-growing bacteria will be more diffusive.

Response:

We used MSDs of bacteria poles to show that they have either symmetrical or asymmetrical adhesion while we used MSDs of the bacteria centers to determine their growth rate. Therefore there is no indirect correlation between the division and symmetrical nature of this division.

The exponent of the MSD has been used to distinguish swimming bacteria, which exhibit a ballistic behavior, diffusing bacteria, which show a diffusive behavior, and attached bacteria, which have a sub-diffusive behavior (Visser et al. 2018). In the present study, the considered dividing and non-dividing bacteria are all attached to the surface. Hence, they generally show sub-diffusive dynamics for both classes and the exponent of the MSD growth law does not provide a clear criteria to distinguish them. **We have added further explanations about this in the revised manuscript (see Methods - “Mean square displacement”, page 18 in the annotated manuscript).**

Reference:

Vissers, T., Brown, A. T., Koumakis, N., Dawson, A., Hermes, M., Schwarz-Linek, J., ... & Poon, W. C. (2018). Bacteria as living patchy colloids: Phenotypic heterogeneity in surface adhesion. *Science Advances*, 4(4), eaao1170.

5. The role of bacterial fimbriae is barely mentioned in the manuscript, although they have been shown to be sensitive to shear in the shear rate regime used in this study. The authors should consider conducting control experiments using a mutant of fimbriae production that eliminates the mechanosensitive effects of bacterial adhesion.

We agree with the referee that fimbriae in *Escherichia coli* play a major role in surface colonization by allowing a mechanical feedback on bacterial adhesion. Increasing shear has been shown to induce conformational changes in the structure of the FimH domain of type 1 fimbriae, thereby giving rise to a catch bonding behavior that strengthens cell adhesion. Nevertheless, the use of a mutant of fimbriae production would pose major challenges in the context of our experiments. It has been shown that type 1 fimbriae are necessary for irreversible adhesion of *E. coli* to surfaces (e.g. Pratt and Kolter 1998, Cookson et al. 2002, Beloin et al. 2008, Wang et al. 2018). Irreversible adhesion is needed for *E. coli* to duplicate on surfaces, and therefore mutants lacking type 1 fimbriae do not grow on surfaces so that their ability to form biofilm is considerably reduced (Niba et al. 2007, Wang et al. 2018). Therefore we expect that experiments with fimbriae mutants would have too broad consequences on bacterial cells, which would not make it possible to test the effect of the shear on the growth rate. The aim of our study is to demonstrate that shear affects the phenotypic heterogeneity in a population of attached bacterial cells. The investigation of the genetic systems and molecular mechanisms by which bacteria sense shear is beyond the scope of the study. **In the revised manuscript, we discuss in more detail the possible role of fimbriae in mechanosensing (page 13 in the annotated manuscript).**

References:

Beloin, C., Roux, A., & Ghigo, J. M. (2008). *Escherichia coli* biofilms. *Bacterial biofilms*, 249-289.

Cookson, A. L., Cooley, W. A., & Woodward, M. J. (2002). The role of type 1 and curli fimbriae of Shiga toxin-producing *Escherichia coli* in adherence to abiotic surfaces. *International Journal of Medical Microbiology*, 292(3-4), 195-205.

Dufrêne, Y. F., & Persat, A. (2020). Mechanomicrobiology: how bacteria sense and respond to forces. *Nature Reviews Microbiology*, 18(4), 227-240.

Pratt, L. A., & Kolter, R. (1998). Genetic analysis of *Escherichia coli* biofilm formation: roles of flagella, motility, chemotaxis and type I pili. *Molecular microbiology*, 30(2), 285-293.

Thomas, W. (2008). Catch bonds in adhesion. *Annu. Rev. Biomed. Eng.*, 10, 39-57.

Wang, L., Keatch, R., Zhao, Q., Wright, J. A., Bryant, C. E., Redmann, A. L., & Terentjev, E. M. (2018). Influence of type I fimbriae and fluid shear stress on bacterial behavior and multicellular architecture of early *Escherichia coli* biofilms at single-cell resolution. *Applied and environmental microbiology*, 84(6).

6. It seems that only one biological replicate has been done for Fig 1 and 2.

Response:

In the original manuscript, the growth curves were plotted from a single experiment. In response to this comment, we have run a series of new experiments allowing us to define confidence intervals from independent triplicate experiments in Figures 2, 4b and 4c. These new experiments confirm our original curves with relatively small confidence intervals. **We have clarified this in the revised manuscript in the paragraph “Shear induces heterogeneous division rates in an isogenic bacteria population” (page 2 in the annotated manuscript) and Methods section (“Bacterial strain and culture conditions”, page 15 in the annotated manuscript).**

7. The method of calculating the effective growth rate η_{eff} is not standard. Could the authors measure more direct properties such as cell elongation rate or total biomass growth rate by fitting exponentials?

Response:

The growth rate is calculated as:

$$\eta = \frac{1}{n} \frac{dn}{dt} \quad (1)$$

This is equivalent as fitting an exponential to the number of bacteria n with

$$n = e^{\eta t} \quad (2)$$

We have verified this by comparing the growth rates estimated from the high acquisition rate measurements using equation 1 (Fig. 3c) to the exponential growth curves of Fig. 2 and 4b using equation 2, showing a good agreement.

We have also measured the statistics of elongation rate of dividing bacteria at different times and for different flow regimes (see figure R5 below). The statistics are similar at all times and in all regimes, with an average elongation rate of approximately $0.05 \mu\text{m} \cdot \text{min}^{-1}$. Normalizing by the bacteria size ($2 \mu\text{m}$) leads to a division rate of 0.025 min^{-1} , equivalent to a mean division time of 40 minutes.

Figure R5: Statistics of elongation rates measured at different times and for different flow regimes.

Minor comments:

1. The term “generation” used in the manuscript is misleading. Figure 1 does not show the map of bacterial generations but rather the time when new tracks appear. Thus, a bacterium that has not yet divided during the 320 first minutes of the experiment but divides after 320 min will be labeled in red even if this is only the second generation.

Response:

We agree with the referee that the term “generation” was ill-chosen. Following the reviewer’s advice, we have removed the term “generation” and replaced it by “bacteria classes tagged according to the time at which their last division occurred”.

2. Could the authors provide the raw movies of the experiments presented in Fig1?

Response:

Following the reviewer’s advice, we have included the raw movies for different flow rates as supplementary videos.

3. The title of figure 1 is misleading because the color code does not directly reflect the measure of the division rate. Even if fewer bacteria divide at the beginning of the experiment, it is not clear if they divide more slowly.

Response:

Following the reviewer’s advice, we have modified the title of the figure to: “Flow induces heterogeneous bacteria lifetimes between successive division events in bacterial colonies on surfaces”.

4. On page 4, it is stated that “the heterogeneity in the temporal distribution of division events increased significantly” but there is no quantitative data to support this statement. Could the authors calculate the variance of these distributions?

Response:

The increase in the heterogeneity of division times refers here to the appearance of a sub-population of non-dividing bacteria when increasing the shear rate. This is demonstrated quantitatively in Figures 4b and 4c by calculating the fraction of non-dividing bacteria for different shear rates. Calculating the variance of the division time distribution would not characterize this phenomenon as it does not include non-dividing bacteria. In fact, as shown in Figure 5 and discussed in the manuscript, the distribution of the division times of dividing bacteria does not depend much on the shear rate. **We have clarified this point in the revised manuscript in the paragraph “The division statistics of dividing bacteria do not depend on flow” (page 9-10 in the annotated manuscript).**

5. On page 4, it is written that “the population growth followed a regular exponential trend (Fig2a)”. This is not what I can see in Fig2a. Moreover, it would be clearer to plot these graphs in log-in scale rather than lin-lin.

Response:

Following the reviewer’s advice, we have plotted the population growth curves in semilog in Figures 2 and 4b. The growth is approximately exponential for most of the observation times in all regimes. This exponential trend develops after an initial lag time needed for lagged dividers to start dividing as discussed in the manuscript. In the case of the ultra low shear experiment, a significant reattachment of bacteria from upstream regions occurs after about 280 minutes, causing an acceleration of the population growth at late times. This is due to the formation of second layers of bacteria in upstream regions, which favors detachment. This only influences the late time growth of the lowest shear experiment and does not affect our general analysis and conclusions. **We have explained this in the revised manuscript in the paragraph “Shear induced by flow can prevent bacteria from dividing” (page 5-6 in the annotated manuscript).**

6. The title of figure 2 is misleading, What is plotted are the growth curves, but not the growth rates calculated from exponential fits to these curves.

Response:

We have modified the title of this figure to: “Flow slows down the growth of bacteria populations on surfaces”.

7. Could the author explain in the Methods how detachment and attachment events are computed? On the supplementary movies, it is clear that the images are imperfectly segmented and some bacteria disappear and reappear in successive frames. Could they also indicate these values in terms of events per bacteria and per unit of time?

Response:

The detachment and attachment events are measured from high acquisition frequency measurements (10 fps). The particles are tracked in time and whether they detach or attach is inferred from the time evolution of the distance between successive positions. There is indeed a certain amount of “flickering”, that is, some particles are not visible at certain times. However, the *Trackpy* algorithm is able to reconstruct a trajectory even when a particle is not visible in several successive time frames within the trajectory, thanks to its prediction framework. Hence this does not affect the detection of attachment/detachment events. **We have expanded the Methods section (“Detection of detachment and attachment events”, page 17-18 in the annotated manuscript) to explain this.** The events of detachment and attachment per bacteria and per unit time are given in Figure 3c.

8. A question somewhat related to minor point 6 concerns the details of the tracking parameters. Could the authors indicate in the Methods the distance that particles are allowed to move between two successive frames and whether the algorithm tolerates discontinuous trajectories in which bacteria disappear for one or more frames? In addition, the authors mentioned in the Methods that the length of a track reflects the division time. This is only the case if and only if there is no error on the track.

Response:

Yes, the algorithm tolerates the disappearances of a particle from several time frames along its trajectory (Allan et al. 2015). The length of the track reflects the division time only if the bacteria is classified as a divider. This classification is based on the value of the MSD at the end of the trajectory. Hence, a loss of trajectory would not lead to an erroneous detection of a division event. **This is explained in more details in the revised manuscript in the Methods section (“Mean Square Displacement”, page 18 in the annotated manuscript).**

9. In figure 3 it would be more informative to use 2D scatter plots rather than bar plots. Indeed, since the value of the shear stress (or shear rate) is known for each condition, it is possible to plot the different rates as a function of the shear stress (or rate).

Response:

For consistency with the other figures in the manuscript, we have kept the original labeling with the four flow regimes. **To make it more straightforward for readers to associate them to a precise value of shear stress or shear rate, we have included a new table (table 1) and added a reference in all figure captions.**

10. On page 6, it is written that “MSD on the order of μm^2 ”. Since MSDs are a function of time, could the author indicate that this value corresponds to the average MSD at division?

Response:

Yes, we have specified this in the revised manuscript in the Methods section (“Mean Square Displacement”, page 18 in the annotated manuscript).

11. In the discussion, at the end of page 10, there are tons of scenarii other than stochasticity in gene expression that can account for heterogeneity.

Response:

We agree with the reviewer and we have removed the reference to stochasticity in gene expression as this is indeed only one scenario amongst others.

12. The VBNC state is very different from the lag phase that bacteria experience when they resume from stationary to exponential phase. If bacteria are non-cultivable it is because we are unable to provide the right environment and sets of interactions for the bacteria to grow in a laboratory.

Response:

Yes, the VBNC state that can be induced by various stresses, and the lag phase, which occurs when bacterial growth resumes, correspond to distinct physiological states. As explained in our answers above, bacteria were actively dividing in our microfluidic experiment. Thus, the growth arrest that we observed for a fraction of the bacterial population is a direct response to the stress induced by an increasing shear, analogous to a VBNC state. It is not a lag phase induced by an effect of the aging of the stock culture or the use of inadequate culture conditions. We also agree that, for a large part of environmental microorganisms, cultivation in the laboratory remains impossible, or at least very challenging. However, this is not the case of *Escherichia coli*.

13. In the Methods, there is a typo in the M9 recipe. Replace NCl by NH₄Cl.

Response:

Corrected.

14. Could the authors indicate in the Methods the time interval between frames for long experiments?

Response:

The time interval between successive frames for long experiments is 1 min (see Methods section “Image acquisition”).

15. Equation 5 should be written in its discrete form.

Response:

We have modified the equation in discrete form instead of its continuum form.

Reviewer #3 (Remarks to the Author):

This is an interesting work which shows the response of *E. coli* cells to different flow strengths in terms of individual growth rates. Much effort has been devoted to image acquisition and image processing in order to identify heterogeneities in the behavior of *E. coli* cells on the surface which may unveil previously unrecognized effects in bacterial surface colonization under fluid flow. However, I see some critical points that should be addressed by the authors.

Response:

We thank the reviewer for his interest in our study and his constructive criticisms.

First of all, the role of type 1 fimbriae - known to play a crucial role in the adhesion process to surfaces in *E. coli* - should be considered not only in the discussion section but also in terms of additional experiments with commonly available type 1 fimbriae mutant strains to confirm or reject the hypothesis that enhanced surface attachment and growth arrest are the two sides of bacterial response to flow.

Response:

Type 1 fimbriae play a major role in regulating bacteria adhesion and surface colonization (Thomas et al. 2008; Dufrene and Persat 2020). The use of type 1 fimbriae mutants to better understand the response to flow-induced mechanical stress is a question worth asking. However, it poses major issues that are prohibitive in the context of our experiments. It has been shown that type 1 fimbriae are necessary for irreversible adhesion of *E. coli* to surfaces (e.g. Pratt and Kolter 1998, Cookson et al. 2002, Beloin et al. 2008, Wang et al. 2018). Irreversible adhesion is needed for *E. coli* to duplicate on surfaces, and thereby mutants lacking type 1 fimbriae do not grow on surfaces and their ability to form biofilm is considerably reduced (Niba et al. 2007, Wang et al. 2018). In addition, investigating the genetic systems and molecular mechanisms by which bacteria sense and respond to shear is certainly an interesting perspective but it is beyond the scope of our study.

In the revised manuscript, we have discussed in more detail the possible role of fimbriae in combined attachment/growth arrest response of bacteria exposed to flow (page 13 in the annotated manuscript).

References:

- Beloin, C., Roux, A., & Ghigo, J. M. (2008). *Escherichia coli* biofilms. *Bacterial biofilms*, 249-289.
- Cookson, A. L., Cooley, W. A., & Woodward, M. J. (2002). The role of type 1 and curli fimbriae of Shiga toxin-producing *Escherichia coli* in adherence to abiotic surfaces. *International Journal of Medical Microbiology*, 292(3-4), 195-205.
- Dufrêne, Y. F., & Persat, A. (2020). Mechanomicrobiology: how bacteria sense and respond to forces. *Nature Reviews Microbiology*, 18(4), 227-240.
- Pratt, L. A., & Kolter, R. (1998). Genetic analysis of *Escherichia coli* biofilm formation: roles of flagella, motility, chemotaxis and type I pili. *Molecular microbiology*, 30(2), 285-293.
- Thomas, W. (2008). Catch bonds in adhesion. *Annu. Rev. Biomed. Eng.*, 10, 39-57.
- Wang, L., Keatch, R., Zhao, Q., Wright, J. A., Bryant, C. E., Redmann, A. L., & Terentjev, E. M. (2018). Influence of type I fimbriae and fluid shear stress on bacterial behavior and multicellular architecture of early *Escherichia coli* biofilms at single-cell resolution. *Applied and environmental microbiology*, 84(6).

As far as the heterogeneity is concerned, it would be interesting to see the effect of a rapid drop or a rapid increase of shear stress on the growth of bacteria on the surface: would non-dividers cells become dividers or vice versa?

Response:

In response to this comment, we have included the results of an additional experiment where the shear rate was changed from high to ultra low after about four hours (Figure R6 below). Results show that the decrease of the shear rate induces a large increase of the population growth rate, consistent with the

trends observed in steady state flows. Bacteria hence appear to adapt rapidly to a change in flow conditions, after a lag time that may be similar to that observed for lagged-dividers. Note that this enhanced colonization rate is partly due to the settling of new bacteria on the surface due to the sharp reduction of flow. The precise investigation of the response of bacteria to flow fluctuations requires a dedicated study, which is a promising perspective of this article. While this is beyond the scope of this article, we **have included this additional experiment in the revised manuscript (Figure SI.5) and discussed the potential for future studies on flow fluctuations (page 13-14 in the annotated manuscript).**

Figure R6: Bacterial growth response to a sudden change in the imposed shear. Number of bacteria attached to the surface as a function of time. The flow rate is changed from high shear to ultra low shear after 260 minutes (red line).

Another crucial point is related to the statistical analysis: it is not clear to me if standard deviations correspond to population variability for a single experiment or if several experiments (i.e. different bacterial suspensions, different channels, etc.) have been performed.

Response:

In the original manuscript, the error bars were calculated from different images of a single experiment. In response to this comment, we have included a series of new experiments allowing us to define confidence intervals from the standard deviation computed over independent triplicate experiments. These new experiments confirm our original curves with relatively small confidence intervals. **We have clarified this in the revised manuscript in the paragraph “Shear induces heterogeneous division rates in an isogenic bacteria population” (page 2 in the annotated manuscript) and Methods section (“Bacterial strain and culture conditions”, page 15 in the annotated manuscript).**

Figure 3: data reported in the text for the observed and effective growth rates for the med and high regimes (0.4-0.48 and 0.25-0.35) do not correspond to the mean values shown in the graph.

Response:

Thank you for pointing out this typo. The correct observed/effective growth rates are respectively 0.35/0.41 for the med regime and 0.21/0.29 for the high regime. **We have corrected these values in the revised manuscript (page 8 in the annotated manuscript).**

Reviewer #1 (Remarks to the Author):

The revised manuscript addressed all of my questions adequately. The revised methods section will benefit researchers wishing to pursue similar investigations.

Reviewer #1 (Remarks on code availability):

I was able to verify execution of the python code. Note that to execute on a Windows OS machine the PATH_DATA variable assignment should be replaced with

```
PATH_DATA = '..\\step1_matlab\\images_treated_20201121_124215\\' # windows
```

Reviewer #2 (Remarks to the Author):

In their revised manuscript, the author have repeated the experiments to obtain biological replicas of the experiments described in the initial submission. They have also better detailed the methods. However, inaccuracies remain in some of the key values used to support the claims put forward in the abstract. I also remain highly skeptical of the conclusions presented in this study, as the raw images are of poor quality and image analysis is handled with a very basic approach (global thresholding).

Major comments:

1. Growth rate measurements are central to demonstrate that higher shear rates do indeed reduce surface colonisation capacity by producing non-dividing bacteria. I struggled to find a precise definition of the observed growth rate. As I understand it, growth rates are measured as the number of divisions per bacterium per unit of time. It is therefore written as $N_{\text{division}}/N_{\text{bacteria}}/\Delta t$, where Δt is the observation time. However, N_{bacteria} varies over the course of the measurement, so this definition is ill-posed. The authors should consider calculating the average growth rate by quantifying the elongation rate obtained from the exponential fit of cell length versus time for individual cells in the population. The authors should therefore provide a supplementary figure with a set of individual traces and the associated fits. The fitting function should a priori be exponential since the insertion of new cell wall is proportional to the cell length. Intriguingly, the authors reported in the manuscript a constant velocity to describe cell elongation. Furthermore, from the curves presented in Figure SI.4, it appears that single cell length data are very noisy. If measurements on individual cells are too difficult, I suggest to do the analysis on the time evolution of the total area occupied by bacteria on a frame.

2. To really establish that the fraction non-dividing cells increases with shear, we need to know how the tracking algorithm performs if it never detects one of the daughter cells, which was, for example, directly advected by the flow after septation. Furthermore, despite the fact that attachment rate is almost zero in high shear regime, it appears in the series of images presented in Figure SI.3 that the number of non-dividing cells increases significantly between 120 min and 240 min at locations where there were no dividing cells. How does this happen? I strongly recommend to plot the total surface occupied by bacteria in these experiments to have a more direct measurement of the efficiency of surface colonisation and fit it with an exponential.

3. There appears to be a discrepancy between Figure 3a and Figure 3c. Although the detachment ratio increases steadily with increasing shear rate in Figure 3a, the detachment rate in the high shear regime is somewhat lower than in the median regime. If measurements in Figure 3a are performed on the exact same time interval, the data in Figure 3a,b should be identical to those reported in Figure 3c. What did I miss?

4. Figure 4b shows data that do not agree with the model proposed by the authors. According to the abstract, the proportion of non-dividing cells should increase with the applied shear rate (also

depicted in the schematic of Figure 4d). However, the population of non-dividing cells is stationary at high shear rate, whereas it increased at lower shear rates. These data do not seem to correspond to the scenario described in the abstract.

5. The model shown in Figure 6d is somewhat oversold. The authors have never measured adhesion. MSD measurements cannot distinguish between bacteria that are not growing and those that are attached at both pole, since the bacterial poles would remain fixed in both cases.

Minor comments:

1. Figure 4c: indicate the unit (h^{-1})
2. Figure 6: The y label in panel a, b and c mentions "absolute", while caption mentions "relative".
3. Typo: "thave" instead of have page 17.

Reviewer #3 (Remarks to the Author):

The authors' response and revisions have satisfactorily addressed my comments on the earlier version of the manuscript.

Response to Reviewer 2

Major comments:

1. Growth rate measurements are central to demonstrate that higher shear rates do indeed reduce surface colonisation capacity by producing non-dividing bacteria. I struggled to find a precise definition of the observed growth rate. As I understand it, growth rates are measured as the number of divisions per bacterium per unit of time. It is therefore written as $N_{\text{division}}/N_{\text{bacteria}}/\Delta t$, where Δt is the observation time. However, N_{bacteria} varies over the course of the measurement, so this definition is ill-posed. The authors should consider calculating the average growth rate by quantifying the elongation rate obtained from the exponential fit of cell length versus time for individual cells in the population. The authors should therefore provide a supplementary figure with a set of individual traces and the associated fits. The fitting function should a priori be exponential since the insertion of new cell wall is proportional to the cell length. Intriguingly, the authors reported in the manuscript a constant velocity to describe cell elongation. Furthermore, from the curves presented in Figure SI. 4, it appears that single cell length data are very noisy. If measurements on individual cells are too difficult, I suggest to do the analysis on the time evolution of the total area occupied by bacteria on a frame.

Since we investigate the effect of flow on the effective growth rate at the population scale, our growth rate measurements are based on the evolution of the number of bacteria in time and not on the measurement of single cell lengths. The latter are relatively noisy for attached bacteria under flow. Furthermore, they are not representative of population scale dynamics.

Hence, as explained in our previous revision, the growth rate is calculated as:

$$\eta = \frac{1}{N} \frac{dN}{dt} \quad (1)$$

where N is the number of attached bacteria. Since we perform our analysis in conditions where bacteria form a single layer on the surface, N is proportional to the total area occupied by bacteria on a frame, which is the metrics suggested by the reviewer. Equation (1) implies :

$$N = e^{\eta t} \quad (2)$$

Therefore, it is equivalent to the logarithmic growth rate obtained by fitting an exponential to the temporal evolution of the number of bacteria N (or occupied area).

By correcting for the effect of detachment/attachment, we obtain the effective growth rate μ_{eff} , which decreases significantly with the shear rate (Fig. 3c). The correlation with the fraction of dividing bacteria (Fig. 4C) and the match of the data with the mathematical model relating the growth rate to the fraction of dividing bacteria (equation 4) demonstrate that higher shear rates do indeed reduce surface colonization capacity by producing non-dividing bacteria.

We have revised the manuscript to clarify this point and added the new Equation (1) to define the growth rate on page 3.

2. To really establish that the fraction of non-dividing cells increases with shear, we need to know how the tracking algorithm performs if it never detects one of the daughter cells, which was, for example, directly advected by the flow after septation. Furthermore, despite the fact that attachment rate is almost zero in high shear regime, it appears in the series of images presented in Figure SI.3 that the number of non-dividing cells increases significantly between 120 min and 240 min at locations where there were no dividing cells. How does this happen? I strongly recommend to

plot the total surface occupied by bacteria in these experiments to have a more direct measurement of the efficiency of surface colonisation and fit it with an exponential.

For any bacterial trajectory, the tracking algorithm ends the trajectory at the time for which it is not able to unambiguously associate the bacterium's position in the previous image to a bacterium in the current image. This occurs either when a cell divides or when it detaches. As explained in the manuscript, we differentiate these two types of events based on the value of the MSDs at the end of the trajectory.

In the images of Figure SI.3, some bacteria indeed appear at locations where there was no bacteria before. This is consistent with the reattachment rate of Figure 3.b that is small for the high shear regime, but not zero: about 10 % of attached bacteria reattach downstream.

As mentioned above, the total surface occupied by the bacteria is proportional to the number of bacteria that we measure on the image. So the data of Fig. 2 is exactly proportional to the total surface occupied by bacteria. We did not fit an exponential to that data, but we show on the figure exponential trends corresponding to the apparent growth rates obtained from the high frequency acquisition experiments; they appear as linear trends in the lin-log plots, and are consistent with the data. Hence, fitting an exponential to the data would provide a similar value. Furthermore, this shows that the growth data from the two types of experiments (low acquisition frequency and high acquisition frequency) are consistent with each other.

We have explained this in the revised manuscript in the first paragraph on the Results section (end of page 3, beginning of page 4).

3. There appears to be a discrepancy between Figure 3a and Figure 3c. Although the detachment ratio increases steadily with increasing shear rate in Figure 3a, the detachment rate in the high shear regime is somewhat lower than in the median regime. If measurements in Figure 3a are performed on the exact same time interval, the data in Figure 3a,b should be identical to those reported in Figure 3c. What did I miss?

The measurements of Figure 3a and Figure 3c are indeed taken on the same time interval and from the same dataset. However, they represent different metrics.

The ratio (non-dimensional) shown in Fig. 3a is defined as:

$$R = \frac{N_d(T)}{N(T)} \quad (3)$$

with $N_d(T)$ the total number of detached bacteria during the observation time T and $N(T)$ the total number of bacteria on the surface at time T.

The detachment rate (unit of inverse time) of Fig. 3c is defined as:

$$\left\langle \frac{1}{N(t)} \frac{dN_d(t)}{dt} \right\rangle \quad (4)$$

where the symbols $\langle \rangle$ denote the average of all times t between 0 and T.

The detachment rate tends to increase with the shear rate (Fig. 3c). But it is indeed slightly lower for the high shear than for the medium shear. This is possibly due to the decay in the fraction of dividers in the high shear regime compared to the medium shear regime (Fig. 4.c). Since dividers are attached by one pole only, they are likely more easily detached than non-dividers, who are attached by both poles.

Despite this slightly smaller detachment rate, the detachment ratio is larger for the high shear than the medium shear (Fig. 3.a). For the high shear regime, bacteria from the few dividing colonies are continuously detached but the total number of bacteria on the surface N remains small due to the large fraction of non-dividers and the small reattachment probability (Fig. 3.b). This leads to a large ratio N_d over N . For the medium shear regime, the detachment rate is slightly larger (Fig. 3c) but about half of detached bacteria reattach downstream (Fig. 3.b) and can create new colonies. Hence the number of attached bacteria grows more relatively to the number of detached bacteria, as compared to the high shear regime. This leads to a smaller detachment ratio (Fig. 3.a).

We have clarified the definition of the detachment rate (new equation 1 in the revised manuscript) and detachment ratio (new equation 7 in the revised manuscript) and included the discussion above in the last paragraph of the results section (page 11).

4. Figure 4b shows data that do not agree with the model proposed by the authors. According to the abstract, the proportion of non-dividing cells should increase with the applied shear rate (also depicted in the schematic of Figure 4d). However, the population of non-dividing cells is stationary at high shear rate, whereas it increased at lower shear rates. These data do not seem to correspond to the scenario described in the abstract.

Indeed, the abstract states that the proportion of the non-dividing bacteria increases with shear. This is consistent with our measurements, which show that the fraction of dividers decreases with the applied shear, and therefore the fraction of non dividers increases with shear (Fig 4.c).

As described by the reviewer, the number of non-dividers tends to increase faster in time when decreasing the shear (orange curves on Fig. 4.b), following the trend of the dividing population (blue curves on Fig. 4.d). This is because a fraction of newly produced cells become non-dividers as discussed in the manuscript. However, the ratio of non-dividers over dividers (ratio of orange to blue curves in Fig. 4.b) increases with shear, consistent with our conclusions. Therefore, although the number of non-dividers grows slower in time for the high shear, their fraction in the population of attached bacteria is larger than under smaller shear rates.

We have added a corresponding discussion in the revised manuscript on page 9.

5. The model shown in Figure 6d is somewhat oversold. The authors have never measured adhesion. MSD measurements cannot distinguish between bacteria that are not growing and those that are attached at both poles, since the bacterial poles would remain fixed in both cases.

As explained in our previous revision, we use independent metrics to identify the bacteria that are not growing and to evaluate the symmetry of their attachment:

- Non-growing bacteria are identified from the MSD of the bacteria's centers of mass.
- The symmetry of the bacteria's attachment is characterized by comparing the MSDs of their two poles.

Bacteria that are not growing have smaller MSD of their center of mass than growing bacteria (Fig. SI.2). If they were attached by one pole only, the MSD of the other pole would be significantly larger, approximately twice the MSD of their center of mass. Yet, the MSD of the poles of non-growing bacteria are much closer to each other (Fig. 6.c) compared to the dividers (Fig. 6.a), suggesting a symmetric attachment.

We have emphasized this in the revised manuscript on pages 10 and 11 and noted that our hypothesis linking growth arrest and symmetric attachment should be confirmed by direct measurements of adhesion.

Minor comments:

1. Figure 4c: indicate the unit (h^{-1})

Corrected.

2. Figure 6: The y label in panel a, b and c mentions “absolute”, while caption mentions “relative”.

Corrected.

3. Typo: “thave” instead of have page 17.

Corrected.